# Robust Multi-Agent Reinforcement Learning with Diverse Adversarial Agent Generation and Contrastive Policy Encoding

## Abstract

Multi-agent reinforcement learning (MARL) has emerged as a promising approach for learning coordination policies in multi-agent systems (MAS). However, policies trained by conventional MARL algorithms often overfit to specific team behaviors, limiting their ability to remain robust when faced with teammate failures or adversarial interventions. Such limitations pose significant challenges to the deployment of MARL in real-world applications. To address these issues, we propose a novel co-evolutionary robust MARL framework that enhances the robustness and generalization of MAS under policy disturbances and adversarial agents. Our framework comprises two key components: (1) **DAAG**: a **D**iverse **A**dversarial **A**gent **G**enerator optimized via an information theoretic objective to produce behaviorally diverse and challenging adversarial agents, and (2) **CAPE**: a **C**ontrastive learning-based **A**gent **P**olicy **E**ncoder that continuously learns informative representations of adversarial agents' policies encountered during training, which are integrated into the MARL agents' policy learning process to enable dynamical adaptation to diverse and evolving adversarial policies. These two components are optimized in a **co-evolutionary training paradigm**, enabling cooperative agents to robustly co-adapt alongside increasingly diverse adversaries. Comprehensive experiments conducted on the Predator-Prey and SMAC benchmarks demonstrate that our framework significantly outperforms baseline methods in both robustness and generalization capabilities.

## 1 Introduction

Multi-agent reinforcement learning (MARL) has demonstrated remarkable potential in enabling agents to learn coordinated behaviors in complex multi-agent systems (MAS), with applications ranging from autonomous driving and collaborative swarm robotics to real-time traffic scheduling (Sun et al., 2025). However, conventional MARL methods tend to induce tightly coupled policies that overfit to fixed teammate behaviors, resulting in implicit coordination patterns (Ning & Xie, 2024). When multi-agent systems encounter abnormal policy perturbations, the policies learned through conventional MARL methods often exhibit significant brittleness, exposing critical weaknesses in their robustness. These issues become especially intractable when disturbances originate from within the MAS, such as teammate failures or agents' malicious behaviors. Furthermore, real-world disturbances are always diverse, unpredictable, and continuously evolving, requiring MARL agents to generalize beyond the specific adversarial conditions experienced during training (Albrecht & Stone, 2018). These factors pose serious challenges for deploying MARL in safety-critical and real-world settings. To overcome these limitations, it is essential to enable MAS to remain robust and adaptive when encountering adversarial attacks or abnormal behaviors from internal agents.

Recent advances in robust MARL have explored various techniques to enhance policy robustness under adversarial perturbations. A prominent research direction formalizes robust policy learning as a zero-sum game between cooperative and adversarial agents, leveraging minimax optimization or adversarial regularization techniques to derive worst-case policies (Li et al., 2019; Nisioti et al., 2021; Jiao & Li, 2024). While these approaches offer theoretical guarantees, they typically rely on static or handcrafted adversaries and thus fail to capture the diversity of real-world disturbances. An alternative approach focuses on injecting perturbations into the environment states or agent observa-

Figure 1: Overall framework of our approach. The left diagram illustrates the training pipeline of the **Diverse Adversarial Agent Generator (DAAG)**. The middle block, **Adversarial Interaction**, illustrates how adversarial agents are randomly sampled and engaged in adversarial training with the multi-agent system. The right diagram shows how the **Contrastive Adversarial Policy Encoder (CAPE)** collaborates with the multi-agent learning framework to enable robust policy learning.

tions to simulate uncertainty, thereby encouraging invariant policies to input-level corruptions (He et al., 2023; Chen et al., 2023; Li et al., 2025). However, these methods often lack explicit mechanisms for controlling behavioral diversity, limiting their ability to simulate variations in adversarial behavior. Evolutionary theory has recently been employed (Yuan et al., 2023) to generate auxiliary adversarial agents, enabling population-based adversarial training that exposes cooperative agents to diverse and challenging adversarial behaviors. While effective, this method incurs a high computational cost due to iterative evolution and lacks an end-to-end learning pipeline, requiring separate optimization for adversarial and cooperative agents.

Moreover, existing methods seldom treat adversarial policy encoding as a core learning problem, limiting their ability to co-adapt with the training dynamics of cooperative agents. In this work, we propose a novel robust MARL framework that enhances policy robustness and generalization under dynamic adversarial agent perturbations. Our contributions can be summarized as follows:

- We propose a **Diverse Adversarial Agent Generator (DAAG)** that learns to generate behaviorally diverse adversarial policies through an information-theoretic objective. This allows MARL agents to learn robust cooperative policies in a broader range of adversarial scenarios.

- We introduce a **Contrastive Agent Policy Encoder (CAPE)** that uses contrastive learning to model adversarial agent policies, while employing a continual learning mechanism to retain the representation ability of previously encountered adversarial behaviors and adapt to new ones.

- We develop a **co-evolutionary training paradigm** that jointly optimizes DAAG, CAPE, and the robust MARL policy. This end-to-end framework enables the continual adaptation of cooperative agents to evolving adversarial agents, thereby enhancing the robustness and generalization of MAS.

## 2 RELATED WORKS

### 2.1 ROBUST REINFORCEMENT LEARNING

Robust reinforcement learning (RRL) aims to train agents that sustain high performance under perturbations in observation Zhang et al. (2021), adversarial actions (Pinto et al., 2017), or environmental uncertainties (Moos et al., 2022; Wang & Zou, 2021). M3DDPG (Li et al., 2019) extends DDPG to a multi-agent zero-sum setting, modeling adversarial perturbations as opponent agents and training robust policies via minimax optimization with continuous adversary updates. RMAQ (He et al., 2023) addresses the challenge of state uncertainty in real-world MARL systems. This study is the first to provide a systematic approach to addressing state uncertainty in MARL. The adversarial domain randomization (ADR) (Chen et al., 2023) method generates environment perturbations through an adversarial generator to enhance policy generalization and adopts prioritized experience replay to accelerate training and enhance policy stability. Our work contributes to this line by integrating

learned adversarial policies and online policy adaptation. For completeness, we provide additional reviews on multi-agent reinforcement learning and contrastive learning in Appendix A.

# 3 PRELIMINARIES

We model the robust cooperative multi-agent reinforcement learning problem as a Decentralized Partially Observable Markov Decision Process (Dec-POMDP) (Oliehoek et al., 2016), defined by a tuple $M = (\mathcal{N}, \mathcal{S}, \{\mathcal{O}^i\}_{i=1}^n, \{\mathcal{A}^i\}_{i=1}^n, \mathcal{T}, r, O, \varphi)$. Here, $\mathcal{N} = \{1, \ldots, n\}$ is the set of agents, $\mathcal{S}$ is the global state space, $\mathcal{A}^i$ and $\mathcal{O}^i$ are the action and observation spaces of agent $i$, $\mathcal{T}(s'|s, \mathbf{a})$ is the transition function, $r(s, \mathbf{a})$ is the shared team reward, $O(o^i|s, i)$ defines the local observation model, and $\varphi \in (0, 1)$ is the discount factor. Due to partial observability, each agent $i$ maintains an action–observation history $\tau_t^i = (o_1^i, a_1^i, \ldots, o_t^i)$ and decentralized stochastic policies operate on histories $\pi_i(a_t^i \mid \tau_t^i)$. Let $\tau_t = (\tau_t^1, \ldots, \tau_t^n)$ denote the joint history. Executing the joint action $\mathbf{a}_t = (a_t^1, \ldots, a_t^n)$ produces a transition $s_{t+1} \sim \mathcal{T}(s_{t+1} \mid s_t, \mathbf{a}_t)$ and team reward $r(s_t, \mathbf{a}_t)$. We adopt the centralized training with decentralized execution (CTDE) paradigm, where global information may be leveraged during training but each agent acts solely based on $\tau_t^i$ at execution time.

**Definition 1: Adversarial-Partner Dec-POMDP.** To explicitly model robustness against unreliable or adversarial teammates, we extend the Dec-POMDP to an *Adversarial-Partner Dec-POMDP (AP-DecPOMDP)*, defined as $\hat{M} = (\mathcal{N}, \mathcal{S}, \{\mathcal{A}^i\}_{i=1}^n, \mathcal{T}, r, \{\mathcal{O}^i\}_{i=1}^n, O, \varphi, \mathcal{Z})$, where $\mathcal{Z}$ denotes a set of adversarial policies. In each episode, one cooperative agent $i \in \mathcal{N}$ may be replaced by an adversarial partner $\pi^{adv} \in \mathcal{Z}$, whose objective is misaligned with the team reward. The resulting joint action is $\hat{\mathbf{a}}_t = (a_t^1, \ldots, a_t^{i-1}, a_t^{adv}, a_t^{i+1}, \ldots, a_t^n)$, which induces a transition $s_{t+1} \sim \mathcal{T}(s_{t+1}|s_t, \hat{\mathbf{a}}_t)$. Let $\pi = (\pi_1, \ldots, \pi_n)$ denote the joint cooperative policy of $n$ agents, and let $\Pi^{adv}$ denote the space of adversarial policies. The robust cooperative policy learning objective is formulated as the following min–max optimization problem:

$$\max_{\pi} \min_{\pi^{adv} \in \Pi^{adv}} \mathbb{E}\left[\sum_{t=0}^{T} \varphi^t \, r(s_t, \mathbf{a}_t, a_t^{adv}) \,\middle|\, a_t^i \sim \pi_i, \ a_t^{adv} \sim \pi^{adv}\right]. \tag{1}$$

This formulation requires the cooperative policy to maintain high expected return even when interacting with diverse and potentially adversarial partners.

**Unsupervised Diverse Policy Generation via Mutual Information:** Unsupervised policy generation aims to learn diverse, temporally consistent behaviors without relying on external rewards or expert supervision. We adopt an unsupervised reinforcement learning approach based on mutual information maximization (Eysenbach et al., 2018), in which a latent style variable $z$ is sampled from a prior distribution and jointly used with the state to condition the policy. In this setting, the policy is expected to produce behaviorally distinct trajectories for different style codes $z$, while maintaining consistency and identifiability among trajectories conditioned on the same code. This consistency-discriminability principle facilitates stable style conditioning and enables fine-grained control over policy diversity. Specifically, we aim to maximize the mutual information between the latent style code $z$ and the agent's trajectory $\tau$ generated by the corresponding policy:

$$\mathcal{I}(z; \tau) = \mathcal{H}(z) - \mathcal{H}(z \mid \tau). \tag{2}$$

Here, $\tau = (s_0, a_0, \ldots, s_T)$ denotes the trajectory sampled from the policy $\pi$, $\mathcal{I}(z; \tau)$ denotes the mutual information between the latent style code $z$ and the trajectory $\tau$, $\mathcal{H}(z)$ is the entropy of the style distribution and $\mathcal{H}(z \mid \tau)$ is the conditional entropy of $z$ given $\tau$ (Barber & Agakov, 2004). Intuitively, $\mathcal{I}(z; \tau)$ quantifies how well one can infer the style $z$ from the trajectory $\tau$, serving as a measure of behavioral distinctiveness. Furthermore, maximizing $\mathcal{I}(z; \tau)$ encourages the generated trajectory to contain as much information as possible about the style variable $z$. Since the conditional entropy $\mathcal{H}(z \mid \tau)$ is generally intractable, we adopt a variational approximation by introducing a posterior estimator $q_\omega(z \mid \tau)$ to approximate the true posterior. This yields a tractable lower bound:

$$\mathcal{I}(z; \tau) \geq \mathbb{E}_{z \sim p(z), \tau \sim \pi(\cdot|z)} \left[\log q_\omega(z \mid \tau) - \log p(z)\right], \tag{3}$$

where $p(z)$ denotes the prior distribution over style codes, typically assumed to be uniform. We treat $z$ as a discrete variable and implement the posterior estimator $q_\omega(z \mid \tau)$ as a multi-class classifier that takes a trajectory $\tau$ as input and outputs a categorical distribution over the latent codes. By maximizing this variational lower bound during training, the policy network $\pi(a_t \mid s_t, z)$ is encouraged to generate behaviorally distinct and identifiable trajectories conditioned on different style codes $z$.

## 4 METHOD

In this section, we present the robust MARL framework for learning resilient coordination policies under adversarial disturbances. We begin by describing the details of the **Diverse Adversarial Agent Generator (DAAG)**; then, we introduce the **Contrastive Agent Policy Encoder (CAPE)**. Finally, we outline the **co-evolutionary training paradigm** that integrates these two components, enabling cooperative agents to effectively learn under evolving and diverse adversarial agents.

### 4.1 DIVERSE ADVERSARIAL AGENT GENERATOR (DAAG)

To enhance the robustness of multi-agent coordination policies in the presence of malicious actions and unexpected failures from internal agents within the MAS, we propose the Diverse Adversarial Agent Generator (DAAG), which learns a set of style-conditioned adversarial policies through reinforcement learning. The overall architecture is illustrated in the left diagram of Figure 1. DAAG optimizes two disentangled reward signals for each adversarial agent:

**Task reward** $r_{task}$: Encourages adversarial behaviors that oppose the cooperative agents' objective. Specifically, adversarial agents are trained to minimize the team reward of the cooperative MAS.

**Style reward** $r_{style}$: Encourages distinguishable behaviors across different style codes, enabling the generation of diverse adversarial policies.

At the beginning of each training episode, a discrete latent style code $z \sim p(z)$ is sampled from a uniform prior and concatenated with the environment state $s_t$ to form an augmented input $s'_t = [s_t; z]$. The adversarial policy $\pi^{adv}(a_t^{adv} \mid s'_t)$ is then conditioned on this input to generate actions, enabling style-controllable adversarial behaviors.

To ensure that the policy generates trajectories that are both diverse across different styles and consistent within the same style, we adopt a mutual information-based objective. As detailed in Section 3, we optimize a variational lower bound of the mutual information $\mathcal{I}(z; \tau)$ using a trainable posterior network $q_\omega(z \mid \tau)$, which is implemented as a multi-class classifier referred to as the *Style Predictor* in Fig. 1. It takes a trajectory $\tau$ as input and outputs a categorical distribution over the latent style codes. Specifically, the style reward in DAAG is given by:

$$r_{style}(s', a^{adv}) = \log q_\omega(z \mid \tau^{adv}) - \log p(z), \tag{4}$$

where $q_\omega(z \mid \tau^{adv})$ is obtained from the classifier output given trajectory $\tau^{adv}$, and $p(z)$ is the prior probability over styles. This design enables direct optimization of the mutual information lower bound, encouraging the policy to generate distinguishable behaviors under different $z$ values.

**Style Predictor Training:** The *Style Predictor* $q_\omega(\cdot)$ is trained jointly with the adversarial policy generator. During each episode, we record the adversarial trajectory $\tau^{adv}$ generated by the policy conditioned on the assigned style code $z$. These $(\tau^{adv}, z)$ pairs are stored in a trajectory buffer and used to supervise the *Style Predictor* via a cross-entropy loss:

$$\mathcal{L}_{style} = -\log q_\omega(z \mid \tau^{adv}), \tag{5}$$

which maximizes the likelihood of the correct code $z$ under the predicted distribution $q_\omega$. This training objective aligns with the variational mutual information bound and ensures that the predictor captures meaningful differences in behavior induced by different style codes.

**Multi-Objective Critic Architecture:** To simultaneously optimize the adversarial agent for both cooperative disruption and behavioral diversity, we adopt a multi-objective critic architecture with two separate value heads (Van Seijen et al., 2017), as illustrated in Figure 1's *policy generator* module. This separation mitigates interference between objectives, leading to more stable optimization and preventing one objective from dominating the learning process. We build our adversarial agent policy generator on the Proximal Policy Optimization (PPO) algorithm (Schulman et al., 2017), which is known for its training stability and empirical performance. At each timestep $t$, the adversarial agent gets an augmented state $s'_t = [s_t; z_i]$, where $z_i \sim p(z)$ is a latent style code indicating the desired behavioral pattern. The actor then selects an action:

$$a_t^{adv} \sim \pi_\theta^{adv}(a_t^{adv} \mid s'_t) = \pi_\theta^{adv}(a_t^{adv} \mid s_t, z_i), \tag{6}$$

which leads to the next state $s_{t+1}$ and generates a task reward $r_t^{task}$ from the environment. In parallel, a style reward $r_t^{style}$ is computed based on Eq. 4.

The critic network consists of two branches: one estimates the task value $V_{\text{task}}(s'_t)$, and the other estimates the style value $V_{\text{style}}(s'_t)$. These branches are trained separately using their respective temporal difference (TD) errors:

$$\delta_t^{task} = R_t^{task} - V_{task}(s'_t), \delta_t^{style} = R_t^{style} - V_{style}(s'_t), \tag{7}$$

where $R_t^{task}$ and $R_t^{style}$ are the accumulated returns used to supervise the task and style value functions. To guide the actor update, the TD errors are combined into a unified advantage function:

$$A_t = \lambda_{task} \cdot \delta_t^{task} + \lambda_{style} \cdot \delta_t^{style}, \tag{8}$$

where $\lambda_{task}$ and $\lambda_{style}$ are coefficients that balance task effectiveness and behavioral diversity. The PPO objective is then computed as:

$$\mathcal{L}_{PPO} = \mathbb{E}_t \big[ \min(r_t(\theta)A_t, \text{clip}(r_t(\theta), 1 - \epsilon, 1 + \epsilon)A_t) \big], \tag{9}$$

where $r_t(\theta) = \frac{\pi_\theta^{adv}(a_t^{adv}|s'_t)}{\pi_{\theta_{\text{old}}}^{adv}(a_t^{adv}|s'_t)}$ denotes the likelihood ratio between the new and old policies, and $\epsilon$ is a hyperparameter that bounds the policy update to ensure training stability.

## 4.2 CONTRASTIVE AGENT POLICY ENCODER (CAPE):

In multi-agent systems, each agent's partial observation is often noisy and limited in information. Relying solely on such partial observations for policy representation learning leads to unstable embeddings that struggle to discriminate between diverse adversarial styles. To enable effective adaptation to diverse and evolving adversarial policies, we propose the *Contrastive Agent Policy Encoder* (CAPE), which learns discriminative representations of adversarial behaviors encountered during multi-agent coordination training. The CAPE consists of two components: a **Local Policy Encoder** and a **Global Policy Encoder**. The *local policy encoder* $f_\psi^{local}$ encodes each cooperative agent $i$'s real-time observation trajectory $\tau^i$ into a local policy embedding $c^i$ as

$$c^i = f_\psi^{local}(\tau^i). \tag{10}$$

The *global policy encoder* $f_\gamma^{global}$ then aggregates these embeddings from all agents via attention pooling to form an global adversarial context representation $c'_t$. On the one hand, the contrastive loss is applied to the global representation, which enforces style discriminability at the team level while back-propagated gradients enhance the quality of each local encoder, encouraging them to learn more stable and style-informative representations. On the other hand, the global context is also provided to the critic, enabling value estimation to explicitly account for adversarial styles and thereby improving the stability and efficiency of robust MARL training. This design allows cooperative agents to dynamically adapt their behaviors according to the adversarial context, with local encoders supporting per-agent adaptive responses and the global policy encoder stabilizing critic learning. Specifically, at time step $t$, each *agent$_i$* uses the *local policy encoder* to extract a local policy embedding $c_t^i$ from its local observation trajectroy $\tau_t^i$, which is concatenated with its local observation $o_t^i$ to form the actor input $[o_t^i; c_t^i]$, producing the action:

$$a_t^i = \pi_\beta^i(o_t^i, f_\psi^{local}(\tau^i)) = \pi_\beta^i(o_t^i, c_t^i). \tag{11}$$

Meanwhile, the *global policy encoder* aggregates the local embeddings across agents to generate a shared context code:

$$c'_t = f_\gamma^{global}(c_t^1, c_t^2, \ldots, c_t^n), \tag{12}$$

which is concatenated with the global state $s_t$, and used for value estimation in the critic network during policy learning. The detailed learning process will be illustrated in Sec. 4.3. This design enables the cooperative agents to dynamically adapt to high-level adversarial contexts, thereby improving robustness under adversarial disturbances. During training, we leverage the supervised contrastive learning objective (Khosla et al., 2020) to structure the policy representation space. Specifically, we sample a batch of observation trajectories $\{\tau_k^i\}_{k=1}^B$ from each agent's replay buffer. For each trajectory $\tau_k^i$, we generate augmented views via random cropping and masking, denoted as $\tau_k^{i'}$ (Ma et al., 2023). Augmented trajectories originating from the same policy style index are treated as positive samples, while those from different styles serve as negative samples. Then, each augmented sample

$\tau_k^{i\,\prime}$ is first processed by the local encoder to obtain individual agent embeddings $c_k^i = f_\psi^{local}(\tau_k^{i\,\prime})$, which are then aggregated by the global encoder to yield a trajectory-level representation:

$$c_k' = f_\gamma^{global}(\{f_\psi^{local}(\tau_k^{i\,\prime})\}_{i=1}^N). \tag{13}$$

Contrastive learning is then applied on the resulting embeddings $\{c_k'\}_{k=1}^B$ to structure the representation space. Augmented trajectories corresponding to the same policy style index are treated as positive samples $c_p'$, while those from different styles serve as negative samples $c_a'$. The contrastive loss for CAPE training is defined as:

$$\mathcal{L}_{contrastive} = \sum_{k=1}^B \frac{1}{|P_k|} \sum_{p \in P_k} -\log \frac{\exp\left(c_k' \cdot c_p'/\sigma\right)}{\sum_{a \notin P_k} \exp\left(c_k' \cdot c_a'/\sigma\right)}, \tag{14}$$

where $P_k$ denotes the set of positive samples sharing the same style index as $k$. The temperature $\sigma$ controls the sharpness of the distribution.

**Continual Contrastive Representation Learning:** To mitigate representation drift and forgetting of previously seen adversarial behaviors during alternating training, we adopt a continual contrastive learning strategy by employing an Instance-wise Relation Distillation (IRD) loss (Cha et al., 2021). Specifically, we maintain a memory buffer $\mathcal{M}$ that stores past training iterations' adversarial trajectories. At each update step, samples from $\mathcal{M}$ are mixed into the current training batch, encouraging the encoder to retain previously learned discriminative representations. Let $\{\tilde{\tau}_k^i\}_{k=1}^B$ denote the observation trajectory sampled from $\mathcal{M}$. We define the instance-wise similarity vector

$$\mathbf{p}(\tilde{\tau}_k^i; \gamma, \psi) = [p_{k,1}, p_{k,2}, ..., p_{k,B}], \tag{15}$$

where $p_{k,j}$ represents the normalized instance-wise cosine similarity between $\tilde{\tau}_k^i$ and $\tilde{\tau}_j^i$'s policy representations. The IRD loss quantifies the discrepancy between the instance-wise similarities of the current representation and the past:

$$\mathcal{L}_{IRD} = \sum_{k=1}^B -\mathbf{p}(\tilde{\tau}_k^i; \gamma^{past}, \psi^{past}) \cdot \log\mathbf{p}(\tilde{\tau}_k^i; \gamma, \psi), \tag{16}$$

where $\gamma^{past}, \psi^{past}$ are the parameters of policy encoders in the past training iterations.

The final objective for policy representation learning is defined as:

$$\mathcal{L}_{rep} = \mathcal{L}_{contrastive} + \kappa \cdot \mathcal{L}_{IRD}, \tag{17}$$

where $\kappa$ is a weighting hyperparameter controlling the strength of temporal consistency regularization. This design ensures that the encoder not only maintains discrimination over a diverse set of adversarial policies but also preserves long-term representation consistency across iterations.

## 4.3 CO-EVOLUTIONARY ROBUST MARL TRAINING

In this subsection, we detail the overall training workflow, where the DAAG, CAPE, and the robust MARL policy are jointly optimized through an alternating, co-evolutionary paradigm. In each training iteration, we employ an alternating two-stage process to train our framework, which we have now clarified in the manuscript. First, we sample a latent style code $z \sim p(z)$ to generate style-conditioned adversarial agents, which are optimized using PPO with a multi-objective critic. The training then alternates between two main stages: (1) DAAG update stage: In this stage, the DAAG adversarial policy generator is trained to produce a diverse set of adversarial behaviors. This phase continues until a certain adversarial training horizon is reached or the policy prediction loss falls below a threshold. During this time, the CAPE module and the cooperative agents remain frozen. (2) CAPE + cooperative policy update stage: After completing the DAAG training stage, we switch to updating the CAPE module and the cooperative MARL policy. During this phase, CAPE encodes the newly generated adversarial styles, and the cooperative agents adapt to these updated adversaries using the learned representations. These two stages repeat in cycles, allowing DAAG to continuously evolve its adversarial population and the cooperative agents to continuously adapt to this evolving distribution with CAPE.

We adopt Multi-Agent PPO (MAPPO) (Yu et al., 2022) as the backbone algorithm for cooperative policy optimization. It is worth noting that our framework is generally compatible with other MARL algorithms that follow the CTDE paradigm. Each cooperative agent receives the concatenated local observation and local policy embedding as input to its actor network:

$$a_t^i = \pi_\beta(o_t^i, c_t^i), \tag{18}$$

where $\pi_\beta$ is a shared policy network across agents. The critic is trained by the following loss:

$$\mathcal{L}_{critic} = \left( Q_\alpha(s_t, \boldsymbol{a}_t, c_t') - R_t \right)^2, \tag{19}$$

where $Q_\alpha$ denotes the joint action-value function and $R_t$ is the team-based return. The actor update follows the same clipped surrogate objective as in Eq. 9.

This co-evolutionary paradigm allows cooperative agents to adapt and coordinate effectively in the presence of evolving adversarial agents and maintain consistent policy embeddings. Meanwhile, the DAAG is progressively enhanced through iterative training, producing more diverse and challenging adversarial agents. The detailed pseudocode is provided in Algorithm 1 in Appendix E.

## 5 EXPERIMENTS

To evaluate the effectiveness of our proposed co-evolutionary robust MARL framework, we conduct experiments in two widely used benchmark scenarios: **Predator–Prey (PP)** (Terry et al., 2021) consists of three predators cooperating to capture a pre-trained prey. In our robustness evaluation, any subset of predators can be replaced by adversarial agents whose objectives deviate from the team reward, allowing us to test robustness under different degrees of internal adversarial interference. **StarCraft Multi-Agent Challenge (SMAC)** (Samvelyan et al., 2019) is also used for evaluation on three maps: 2s3z, 3m, and MMM. Similar to PP, any subset of ally units may act as adversarial agents, enabling robustness assessment across varying adversarial configurations. By training ally team in the presence of such adversarial interference, we assess the ability of our method, which is not only effective in standard cooperative settings but also robust to adversarial perturbations and policy mismatches.

To ensure unbiased evaluation, the adversarial policies used for testing are not generated by DAAG. Instead, we adopt the minimax population-based adversarial policy generation procedure of Vinitsky et al. (2020) to build an independent adversarial population. This providing a set of unseen, distribution-shifted adversarial policies for robustness evaluation. The detail description are provided in Appendix C.2.

### 5.1 BASELINE METHODS

**Vanilla MAPPO**: Our backbone algorithm without any adversarial training component. It represents a standard cooperative multi-agent RL method and serves as the baseline for evaluating robustness improvements.

**RANDOM**: This baseline introduces a teammate that follows a random policy throughout training, serving as an unstructured source of disturbance to team coordination.

**M3DDPG (Li et al., 2019)**: A robust MARL algorithm that models adversarial perturbations as opponent agents in a zero-sum game and trains via a minimax optimization framework. Adversarial agents are continuously updated to approximate worst-case behaviors.

**ROMANCE (Chen et al., 2023)**: A robust MARL method based on evolutionary algorithms, which generates a diverse set of high-performing policies and trains cooperative agents against them.

### 5.2 RESULTS AND ANALYSIS

**Policy Robustness Evaluation:** Fig. 2 presents the evaluation results of our method compared with four baselines on PP and three SMAC scenarios (2s3z, 3m, MMM) when tested against additional unseen adversarial agent policies. In the figures, solid lines denote the mean return or win rate over five runs with different random seeds, and the shaded regions represent the 95% confidence intervals. Vanilla MAPPO struggles in all scenarios due to its lack of robust training mechanisms. Since

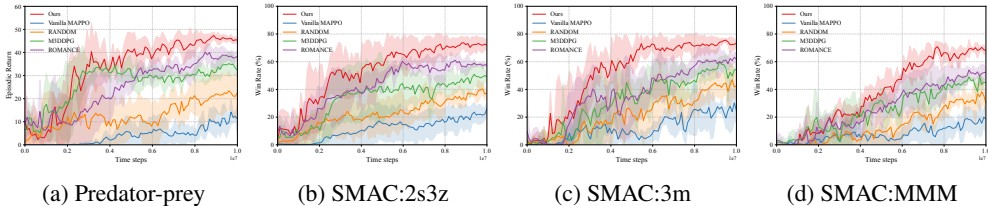

(a) Predator-prey      (b) SMAC:2s3z      (c) SMAC:3m      (d) SMAC:MMM

Figure 2: Average test results in Predator-prey and SMAC scenarios during the training phrase.

RANDOM is unable to produce meaningful adaptation, it results in consistently poor performance, only slightly outperforming Vanilla MAPPO. M3DDPG benefits from adversarial training but tends to overfit to worst-case scenarios, leading to conservative policies that underperform against diverse adversarial behaviors. ROMANCE gains from training with diverse policies but shows limited zero-shot ability when the evaluation adversaries differ from the training. In contrast, our method consistently achieves faster learning and higher final performance across all tasks, indicating that jointly leveraging diverse adversarial behavior generation and contrastive adversarial policy encoder improves robustness to malicious actions in the MAS.

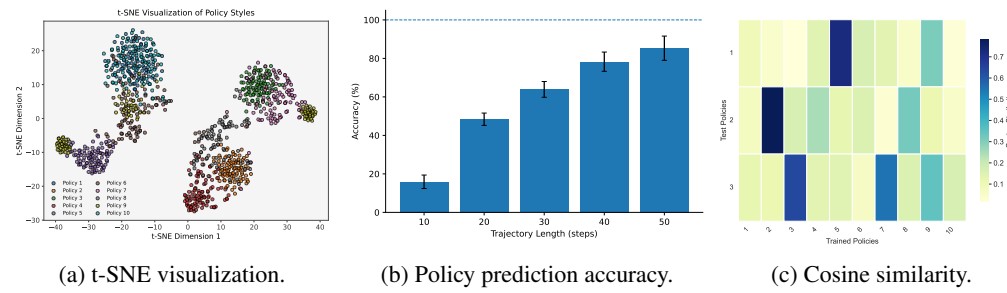

(a) t-SNE visualization.      (b) Policy prediction accuracy.      (c) Cosine similarity.

Figure 3: Comprehensive analysis of learned policy representations through visualization, prediction accuracy, and similarity evaluation. (a) t-SNE visualization of policy embeddings learned from 10 training policies. (b) Policy prediction accuracy for 10 training policies with different trajectory lengths, with error bars denoting 95% confidence intervals over 5 runs. (c) Cosine similarity between test and trained policy representations.

**Agent Generation and Representation Analysis:** To investigate the performance of the DAAG and CAPE, we analyze the learned policy embeddings from CAPE and evaluate their relation to test policies in SMAC's 3m scenario. Fig. 3a shows the t-SNE (Maaten & Hinton, 2008) projection of the learned embeddings for ten distinct policies generated by DAAG during training. Each point in the figure represents a policy representation generated from an observation trajectory of the agent by local encoder, with different colors indicating different adversarial policy agents encountered by the cooperative agents. While most clusters remain distinctly separated, a few exhibit overlap, which may indicate that these policies correspond to similar adversarial policies. Overall, the visualization suggests that CAPE effectively captures discriminative features that distinguish between different policy styles. This property is crucial for enabling MARL to train robust cooperative policies tailored to different adversarial agent behaviors.

To further verify that the local encoder captures distinct adversarial policies, we train a predictor on its embeddings to classify policy categories from trajectories of different lengths. As shown in Figure 3b, the prediction accuracy improves with longer trajectories, indicating that the encoder has indeed learned informative and discriminative policy representations.

Fig. 3c presents the cosine similarity between three unseen test policies and the ten learned training policies. Each cell denotes the average similarity between embeddings from a test policy and those from a training policy. The results show that each test policy tends to have high similarity with one or a small subset of training policies. For instance, *Test Policy1* shows a strong similarity to *Trained Policy5*, while *Test Policy2* exhibits the highest alignment with *Trained Policy2*. These results demonstrate that CAPE learns a discriminative and well-structured representation of diverse

adversarial behaviors, enabling it to effectively map unseen behaviors to semantically related training policies. This capability facilitates robust generalization to novel abnormal agents. Furthermore, the results reveal that DAAG, through alternating iterative training, generates a sufficiently diverse set of adversarial policies that provides broad coverage of the policy space, further supporting the robustness and adaptability of the cooperative policies.

Table 1: Ablation study results. **w/o CAPE** denotes removing the CAPE module from the original architecture. **w/o Multi-Critic** denotes replacing the Multi-Objective Critic in DAAG with a standard actor–critic structure. Best results in **bold**.

| Method | Predator–Prey Score ↑ | 2s3z Win (%) ↑ | 3m Win (%) ↑ | MMM Win (%) ↑ |
|---|---|---|---|---|
| Original | **46.23 ± 1.16** | **73.38 ± 2.42** | **75.46 ± 2.87** | **69.85 ± 3.17** |
| w/o CAPE | 41.25 ± 2.31 | 63.23 ± 2.33 | 58.74 ± 3.63 | 52.25 ± 3.28 |
| w/o Multi-Critic | 36.84 ± 2.76 | 68.79 ± 3.57 | 64.52 ± 4.83 | 57.24 ± 4.19 |

### 5.3 ABLATION STUDY

To assess the contribution of critical components in our framework, we perform an ablation study by systematically removing the Contrastive Adversarial Policy Encoder (CAPE) and the Multi-Objective Critic in DAAG, and measuring their impact on performance across different scenarios. **Effect of CAPE:** To verify the effectiveness of Contrastive Adversarial Policy Encoder (CAPE), we conduct an ablation study by removing CAPE from the architecture. As the result shown in Table 1, the **w/o CAPE** variant's performance drops drastically across all evaluated scenarios compared to the original model. This degradation highlights the importance of CAPE in learning compact and discriminative representations and enabling the cooperative agents to adapt to adversarial policies. **Effect of Multi-Objective Critic:** To evaluate the impact of the Multi-objective Critic in DAAG, we replace it with a standard actor–critic structure. As shown in Table 1, the **w/o Multi-Critic** variant exhibits a notable performance drop across all evaluated scenarios compared to the original model. This degradation demonstrates the importance of multi-objective value estimation for balancing task effectiveness and behavioral diversity, and highlights its role in enabling the cooperative agents to maintain robust cooperation under adversarial conditions. Overall, these results demonstrate that removing either component causes clear and consistent performance degradation. For completeness, we provide additional ablation results in Appendix D.5, which further verify the complementary contributions of all components to robustness under adversarial perturbations.

## 6 CONCLUSION

We propose a robust multi-agent reinforcement learning framework that addresses the challenges posed by diverse and evolving adversarial agents within the MAS. Our method integrates two key components: a Diverse Adversarial Agent Generator (DAAG), which learns style-conditioned adversarial policies through disentangled objectives, and a Contrastive Agent Policy Encoder (CAPE), which encodes adversarial behaviors into compact and discriminative representations. These components are jointly optimized under an alternating co-evolutionary training paradigm, enabling cooperative agents to continually adapt their policies in response to increasingly sophisticated adversaries. Extensive evaluations demonstrate the effectiveness of our approach in improving robustness and generalization under adversarial disturbances. Our framework is broadly compatible with other CTDE-based MARL algorithms and provides a principled foundation for future work in robust multi-agent learning. In future work, we will explore approaches for enhancing the adaptability of our architecture to real-world and open-world scenarios.

## 7 REPRODUCIBILITY STATEMENT

For the details of the model structure and training procedure, please refer to Appendix E. To run our method, please download the code in the supplementary material and follow the instructions in the README files.

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

## A ADDITION REALTED WORKS

### A.1 MULTI-AGENT REINFORCEMENT LEARNING

Multi-agent reinforcement learning (MARL) (Littman, 1994) has remained an active and influential research direction in artificial intelligence for decades. It's a powerful paradigm for learning cooperative policies in environments with multiple interacting agents. Early works such as VDN (Sunehag et al., 2017), COMA (Foerster et al., 2018), and QMIX (Rashid et al., 2020) introduced centralized training with decentralized execution (CTDE) to enable cooperation under partial observability. Extensions such as MADDPG (Lowe et al., 2017) and MAPPO (Yu et al., 2022) further demonstrated strong performance in both cooperative and mixed cooperative-competitive settings. More recently, Wen et al. (2022) reformulated MARL as a sequence modeling problem, enabling the use of Transformer-based architectures to capture temporal dependencies and inter-agent interactions. Despite their successes, most existing MARL methods suffer from overfitting to specific teammate behaviors and fail to generalize to unseen dynamics or agents, especially in the presence of adversarial disruptions (Oroojlooy & Hajinezhad, 2023).

### A.2 CONTRASTIVE LEARNING

Contrastive learning is a machine learning paradigm that acquires representations by comparing instances, pulling semantically related samples closer while pushing apart unrelated ones without relying on explicit labels (Oord et al., 2018). This principle has attracted substantial attention and has been successfully applied in diverse domains, including computer vision (Sung et al., 2024), natural language processing (Rethmeier & Augenstein, 2023), and reinforcement learning (Eysenbach et al., 2022; Zhu et al., 2022). SimCLR (Chen et al., 2020) demonstrated that large-batch training with abundant negative samples can significantly improve representation quality. MoCo (He et al., 2020) introduced a momentum-updated encoder to maintain feature consistency across training iterations. In the reinforcement learning field, contrastive learning has been employed to model the policies of other agents (Ma et al., 2023) or enhance observation representations (Laskin et al., 2020; Zhu et al., 2022). Our approach introduces a contrastive policy encoder that captures the behaviors of adversarial agents, enabling cooperative agents to condition their policies based on informative policy representations. To the best of our knowledge, this is the first work in robust multi-agent learning to apply contrastive learning for adversarial policy modeling.

## B INFORMATION-THEORETIC OBJECTIVE FOR DAAG

The Diverse Adversarial Agent Generator (DAAG) aims to produce a set of adversarial policies that are diverse in behavior, in order to better challenge the cooperative agents. To formalize this objective, we leverage an information-theoretic formulation (Reza, 1994; Barber & Agakov, 2004) based on mutual information (MI) between a latent style variable $z$ and the generated trajectory $\tau$.

### B.1 MUTUAL INFORMATION FORMULATION

Let $z \in \mathcal{Z}$ denote a discrete latent style code sampled from a prior $p(z)$, and let $\tau = (s_0, a_0, s_1, a_1, \ldots, s_T)$ denote the trajectory generated by the adversarial agent's policy $\pi_\theta(a|s, z)$. We aim to maximize the mutual information (MI) between $z$ and $\tau$:

$$I(z; \tau) = H(z) - H(z|\tau), \tag{20}$$

where $H(\cdot)$ denotes entropy. Since $H(z)$ is constant for a uniform prior $p(z)$, maximizing $I(z; \tau)$ reduces to minimizing the conditional entropy:

$$H(z|\tau) = -\mathbb{E}_{p(z,\tau)}\left[\log p(z|\tau)\right]. \tag{21}$$

### B.2 VARIATIONAL LOWER BOUND DERIVATION

Direct computation of $H(z|\tau)$ is intractable for complex policies and environments, since the true posterior $p(z|\tau)$ is unknown. To address this, we introduce a variational distribution $q_\omega(z|\tau)$ parameterized by $\omega$ to approximate the true posterior $p(z|\tau)$. We utilize the non-negativity property of the

Kullback–Leibler (KL) divergence (Cover, 1999):

$$\mathrm{KL}\big(p(z|\tau)\,\|\,q_\omega(z|\tau)\big) \geq 0. \tag{22}$$

By definition, the KL divergence is:

$$\mathrm{KL}\big(p(z|\tau)\,\|\,q_\omega(z|\tau)\big) = \mathbb{E}_{p(z,\tau)}\left[\log\frac{p(z|\tau)}{q_\omega(z|\tau)}\right]. \tag{23}$$

Expanding the logarithm yields:

$$\mathbb{E}_{p(z,\tau)}\left[\log p(z|\tau) - \log q_\omega(z|\tau)\right] \geq 0. \tag{24}$$

Rearranging gives:

$$\mathbb{E}_{p(z,\tau)}\left[\log p(z|\tau)\right] \geq \mathbb{E}_{p(z,\tau)}\left[\log q_\omega(z|\tau)\right]. \tag{25}$$

Therefore, the conditional entropy satisfies:

$$H(z|\tau) \leq -\mathbb{E}_{p(z,\tau)}\left[\log q_\omega(z|\tau)\right]. \tag{26}$$

This inequality provides a tractable lower bound for the mutual information objective:

$$I(z;\tau) = H(z) - H(z|\tau) \geq H(z) + \mathbb{E}_{p(z,\tau)}\left[\log q_\omega(z|\tau)\right]. \tag{27}$$

Since $H(z)$ is constant when $p(z)$ is fixed, maximizing the mutual information is equivalent to:

$$\max_\omega \; \mathbb{E}_{p(z,\tau)}\left[\log q_\omega(z|\tau)\right]. \tag{28}$$

This variational lower bound is tight when $q_\omega(z|\tau) = p(z|\tau)$ almost everywhere.

### B.3 DAAG Objective

Maximizing the lower bound derived from Eq. 28 can be interpreted as a classification objective for predicting the latent style $z$ from the generated trajectory $\tau$. Thus, the final objective for the **Diverse Adversarial Agent Generator (DAAG)** combines the environment task reward $r_{\text{task}}$ and the scaled style reward $r_{\text{style}}$:

$$\mathcal{J}_{\text{DAAG}} = \mathbb{E}\left[r_{\text{task}} + \alpha_{\text{style}} \cdot r_{\text{style}}\right], \tag{29}$$

where

$$r_{\text{style}} = \log q_\omega(z|\tau) - \log p(z), \tag{30}$$

and $\alpha_{\text{style}}$ controls the balance between maximizing environment-specific task reward and encouraging diversity in the generated adversarial policies.

### B.4 Integration with Reinforcement Learning

The final objective for the adversarial agent is a multi-objective PPO loss that combines the environment task reward $r_{\text{task}}$ with the style reward $r_{\text{style}}$. This encourages the adversarial agent to not only maximize its task objective (e.g., catching prey or defeating allies in SMAC) but also to maintain diverse, style-specific behaviors across episodes.

## C Experiment Details

### C.1 Environments

**Predator-Prey (PP).** The predator-prey task is based on the continuous Multi-Agent Particle Environment (MPE). The environment consists of two fixed obstacles, three predator agents and one prey agent moving in a two-dimensional continuous space. As shown in Fig. 4a. Agents observe relative positions and velocities of nearby entities within their field of view, and select movement actions (e.g., four directions and stay still) at each timestep. The prey follows a fixed, pre-trained evasive policy, while the predators aim to coordinate in order to capture the prey within a given horizon. Performance is measured by the capture success rate and episode rewards, averaged over multiple seeds.

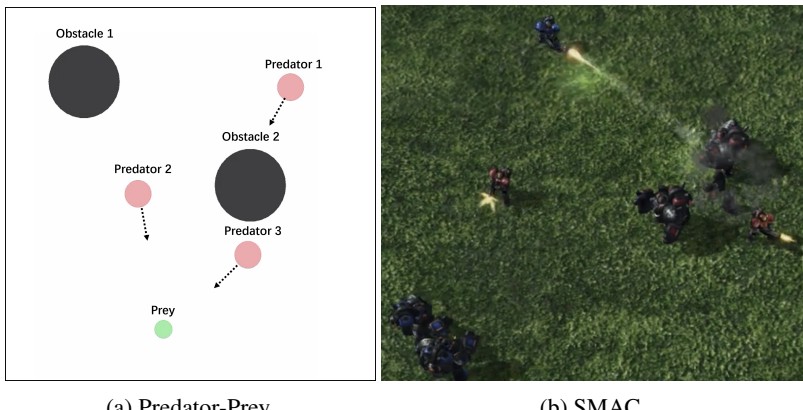

(a) Predator-Prey.                                (b) SMAC.

Figure 4: Evaluation environments: (a) Predator-Prey; (b) SMAC.

**StarCraft Multi-Agent Challenge (SMAC).** The StarCraft Multi-Agent Challenge (SMAC) is a benchmark built upon the real-time strategy game StarCraft II, see Fig. 4b. In SMAC, multiple allied units must collaborate to defeat enemy units under partial observability and complex combat dynamics. It has become one of the most widely used testbeds for evaluating multi-agent reinforcement learning algorithms in challenging discrete-action environments. Each map defines a team of allied units controlled by cooperative agents, which must defeat an enemy team of equal size and composition. Observations consist of local features within a sight range (including ally/enemy states and distances), while actions include discrete movement and attack.

**Adversarial Agent Definition:** In our approach, adversarial behavior is modeled by replacing one cooperative agent with an adversarial agent in each episode. Specifically, at the beginning of every episode, we randomly select a cooperative agent and substitute its policy with an adversarial policy sampled from a pre-defined pool of trained adversarial policies. The adversarial agent follows a stationary policy and executes it throughout the entire episode, thereby simulating persistent adversarial behavior. This design captures the threat scenario of an "insider" agent being compromised or behaving maliciously. Unless otherwise specified, only one cooperative agent is replaced per episode, though the formulation can in principle be extended to multiple adversarial agents.

**Evaluation Protocol:** During testing, we construct an adversarial policy pool independent of training. In each episode, an adversarial policy is randomly sampled from this pool and assigned to the replaced cooperative agent. The cooperative team then interacts under this adversarial disruption, and the performance is recorded. All results are averaged over multiple random seeds and episodes to account for stochasticity. This setup ensures that the evaluation reflects robustness to unseen adversarial policies, rather than memorization of adversarial behaviors encountered during training.

## C.2   Testing Policy Generation

To ensure that the evaluation of robustness is not biased by the adversaries used during training, we construct an independent set of adversarial policies for testing. Specifically, we adopt the population-based adversarial generation procedure proposed by Vinitsky et al. (2020), in which a population of adversaries is initialized with diverse random seeds and trained via a minimax objective. At each rollout, one adversary is uniformly sampled from the population to interact with the cooperative agents, and its policy is updated using PPO based on the collected trajectories. This procedure naturally yields a diverse set of adversarial behaviors without requiring explicit diversity regularization, as the stochasticity from random initialization and gradient updates encourages adversaries to exploit different failure modes of the cooperative policy.

For evaluation, we freeze the trained adversarial population and use it as the testing adversary set, ensuring that robustness is measured against policies not seen during the cooperative training process.

C.3 COMPUTING RESOURCES

Experiments were carried out on a High-Performance Computing (HPC) cluster configured with dual Intel Xeon Gold 5315Y processors (16 cores in total), two NVIDIA L4 GPUs each with 24GB of VRAM, and 256GB of DDR4 system memory. The software environment was based on Linux, with implementations developed in PyTorch and accelerated using CUDA 12.8. We additionally employed SMAC and PettingZoo (Terry et al., 2021) libraries for multi-agent reinforcement learning environments.

C.4 COMPUTATIONAL COST COMPARISON

To provide a fair comparison of training efficiency across different methods, we report the wall-clock training time required to reach convergence on four evaluation environments. All methods are trained under identical hardware configurations (one NVIDIA A100 GPU) and follow the same number of environment steps and rollout settings.

Table 2 reports the training time (in hours) for MAPPO, M3DDPG, ROMANCE, and our DAAG+CAPE approach. As shown, MAPPO is the fastest baseline, while ROMANCE incurs the highest cost due to repeated evaluations over adversarial populations. M3DDPG is moderately slower than MAPPO as it requires decentralized critics for each agent.

Our method introduces two additional components, DAAG and CAPE, but they are trained in alternating stages rather than simultaneously, resulting in only a modest overhead over MAPPO (approximately $1.30$–$1.40\times$ depending on the environment). Importantly, our method remains substantially more efficient than ROMANCE, which requires population-level minimax rollouts. These results demonstrate that DAAG+CAPE achieves strong robustness while maintaining competitive computational efficiency.

Table 2: Wall-clock training time on four evaluation environments.

| Method | Predator-Prey | 2s3z | 3m | MMM | 1c3s5z |
|--------|---------------|------|------|------|--------|
| MAPPO | 3.4 h | 9.1 h | 10.2 h | 12.3 h | 12.6 h |
| M3DDPG | 3.8 h | 11.9 h | 13.2 h | 14.6 h | 14.7 h |
| ROMANCE | 6.1 h | 16.4 h | 17.5 h | 19.2 h | 19.4 h |
| Ours | 4.4 h | 12.8 h | 15.2 h | 16.8 h | 17.2 h |

# D ADDITIONAL EXPERIMENT

## D.1 PERFORMANCE UNDER DYNAMIC ADVERSARIAL SWITCHING

**Research Question:** How well do different methods maintain cooperative performance when the adversarial partner changes unpredictably during an episode? This experiment evaluates whether learned policies can adapt to rapidly varying adversarial behaviors in a non-stationary test setting.

**Testing Partners:** During evaluation, we introduce *dynamic adversarial switching*. Within a single episode, the adversarial partner is periodically replaced (every 15 timesteps) by a newly sampled unseen adversarial policy drawn from the testing adversarial policy pool. This produces a sequence of abrupt behavioral shifts, forcing the evaluated method to continuously adjust to an evolving adversarial strategy rather than adapting to a fixed opponent. All methods face the same switching schedule to ensure fair comparison.

**Experiment Result:** The results in Fig. 5a show that dynamic adversary changes significantly increase task difficulty across all SMAC scenarios. **Vanilla MAPPO** suffers the most, with win rates dropping below 20% on all maps, reflecting its inability to respond to unexpected deviations in teammate behavior. **RANDOM** performs slightly better but remains highly unstable, confirming

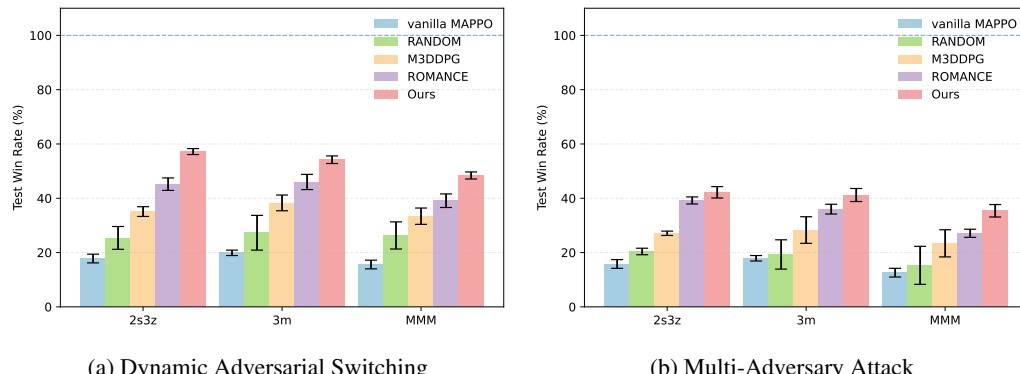

(a) Dynamic Adversarial Switching         (b) Multi-Adversary Attack

Figure 5: Robustness evaluation under challenging adversarial settings. (a) Dynamic Adversarial Switching: the adversarial agent is periodically replaced with unseen adversaries within a single episode. (b) Multi-Adversary Attack: multiple cooperative agents are simultaneously replaced by unseen adversaries.

that the lack of structured modeling makes it ineffective against shifting adversaries. **M3DDPG** exhibits moderate robustness but shows clear performance degradation, especially on complex maps such as MMM, indicating that adversarial training alone is insufficient when adversarial behaviors vary rapidly. **ROMANCE** performs better by leveraging diverse training policies, but its win rate still drops substantially under rapid switches, revealing limited zero-shot adaptability to unseen and quickly changing adversarial strategies.

In contrast, **Our method** consistently achieves the highest win rates across all scenarios. On three testing maps, our approach outperforms ROMANCE by a large margin, demonstrating stable decision-making even when adversarial behavior changes multiple times within an episode. These results indicate that combining diverse adversarial generation with a contrastive adversarial policy encoder enables fast recognition of malicious intent and effective adaptation to dynamic threats.

### D.2 Performance under Multi-Adversary Attack

**Research Question:** Can robust training methods maintain cooperative performance when multiple teammates become adversarial simultaneously? This experiment investigates robustness under increased adversarial influence, where coordination becomes substantially more challenging.

**Testing Partners:** Unlike standard settings where only one agent is replaced during evaluation, here we allow *multiple cooperative agents* to be replaced by adversarial ones at the start of each episode. In particular, we test configurations where two normal agents are substituted with unseen adversarial policies. This creates stronger coordination disruptions and evaluates robustness under high-adversarial-coverage scenarios.

**Experiment Result:** As shown in Fig. 5b, performance decreases for all methods due to the intensified adversarial pressure. **Vanilla MAPPO** nearly collapses in all maps, with win rates approaching zero, highlighting its complete vulnerability when more than one agent behaves maliciously. **RANDOM** also performs poorly, confirming that random behavior models cannot sustain cooperation under multi-agent corruption. **M3DDPG** shows slightly better results but remains highly conservative, resulting in relatively low win rates across maps. This indicates that its worst-case training objective does not generalize when multiple adversaries jointly disrupt coordination. **ROMANCE** performs more competitively, benefiting from having trained against diverse partners, yet it still suffers significant degradation, particularly on MMM, where cooperative structure is more complex and sensitive to adversarial interference.

Our method achieves the strongest performance across all three maps, maintaining substantially higher win rates compared with all baselines. The improvement is most pronounced on 3m and MMM, where adversarial influence is more damaging. These results demonstrate that the proposed

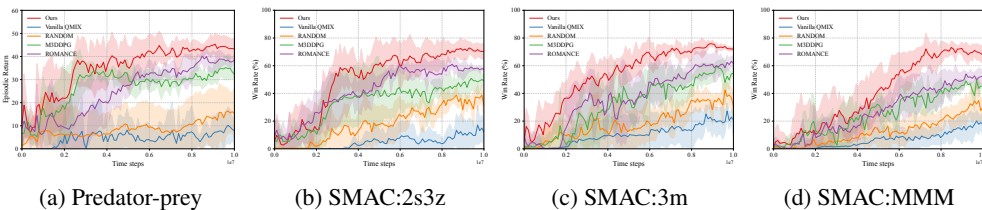

(a) Predator-prey      (b) SMAC:2s3z      (c) SMAC:3m      (d) SMAC:MMM

Figure 6: Evaluation of our framework integrated with the value-based QMIX backbone.

framework not only handles single-adversary disruption but also scales to more severe corruption scenarios involving multiple coordinated adversaries.

### D.3 PERFORMANCE ON VALUE-BASED MARL: QMIX BACKBONE

**Research Question:** While our main experiments employ MAPPO as the MARL backbone, it remains important to verify whether the proposed framework generalizes across different classes of cooperative learning architectures. To assess the backbone-agnostic nature of our approach, we further integrate our adversarial training framework into *QMIX*, a representative and widely adopted value-based MARL algorithm.

**Testing Setup:** We replace the MAPPO backbone with QMIX while keeping all components of our framework unchanged, including the adversarial partner generator and the contrastive adversarial policy encoder. All baselines are reimplemented under the QMIX setting to ensure a fair comparison. Evaluations are conducted in Predator–Prey, 2s3z, 3m, and MMM following the same adversarial testing protocols used in the main experiments.

**Experiment Result:** As shown in Fig. 6, the overall patterns are consistent with our MAPPO-based results. **Vanilla QMIX** suffers severely across all environments, achieving the lowest returns or win rates, which shows that purely value-based factorization methods struggle to cope with adversarial disturbances. **RANDOM** remains unstable and provides poor robustness, while **M3DDPG** performs moderately but becomes overly conservative, leading to suboptimal adaptation to diverse adversarial behaviors. **ROMANCE** achieves better performance by training against multiple partners, but its robustness plateaus and degrades on more complex maps such as 3m and MMM.

In contrast, **our method consistently achieves the best performance across all four tasks**. The advantage is particularly large on 3m and MMM, demonstrating that the proposed framework can effectively enhance adversarial robustness even when paired with a value-based backbone. These results confirm that the core mechanisms of our approach, diverse adversarial behavior generation and contrastive adversarial policy encoding, generalize beyond actor–critic algorithms and remain effective under the QMIX architecture. This experiment verifies that our method is not tied to a specific MARL backbone. Its consistent improvements on QMIX strengthen our claim of generality and demonstrate that the proposed framework can enhance robustness across both value-based and policy-based MARL methods.

### D.4 PERFORMANCE UNDER NORMAL COOPERATIVE SETTINGS

**Research Question:** Do our methods degrade cooperative performance when no adversarial interference occurs? We evaluate multiple approaches under *normal cooperative settings* (i.e., all teammates are aligned and no attack is present).

**Testing Partners:** In this evaluation, *no adversarial agent* is introduced. All agents for testing are trained by cooperative objectives. To evaluate cooperation performance under diverse cooperative settings, we construct testing partners by saving policy checkpoints from different stages of training. Each checkpoint is treated as an independent cooperative partner during evaluation. This ensures that the tested agents interact with partners exhibiting varying levels of training progress and behavioral maturity, rather than only with the final trained model.

**Experiment Result:** As shown in Table 3, the performance of different methods under normal cooperative settings varies significantly. **Vanilla MAPPO** achieves the best results across all en-

vironments. This indicates that in the absence of adversarial interference, a standard cooperative MARL algorithm can fully exploit team coordination.

In contrast, the **RANDOM** baseline performs the worst, with all metrics considerably lower than other methods. For example, it only achieves a return of 24.43 in Predator–Prey, and its win rates in 3m and MMM drop to 49.26% and 43.85%, respectively, showing that random teammate behaviors severely hinder coordination.

**M3DDPG** and **ROMANCE** achieve moderate performance: M3DDPG reaches a return of 43.75 in Predator–Prey but lags behind Vanilla MAPPO on all SMAC tasks; ROMANCE achieves win rates above 80% in 2s3z and MMM but still suffers from noticeable degradation compared with Vanilla MAPPO.

Our method, **Ours**, maintains strong cooperative performance even without adversarial agents. It achieves competitive win rates on 2s3z and 3m, close to the best performance of Vanilla MAPPO, while also showing stable results in Predator–Prey and MMM. Although slightly lower than Vanilla MAPPO in some cases, our approach clearly outperforms M3DDPG and ROMANCE overall.

In summary, these results demonstrate that our method preserves cooperative effectiveness in non-adversarial environments, indicating that robustness improvements are achieved without sacrificing nominal cooperative performance.

Table 3: Average test results of different methods under normal cooperative settings.

| Method | Predator–Prey Return ↑ | 2s3z Win (%) ↑ | 3m Win (%) ↑ | MMM Win (%) ↑ |
|---|---|---|---|---|
| Vanilla MAPPO | **58.75 ± 5.32** | **91.87 ± 2.52** | **89.74 ± 2.65** | **90.43 ± 4.29** |
| RANDOM | 24.43 ± 8.84 | 54.12 ± 3.73 | 49.26 ± 3.54 | 43.85 ± 3.66 |
| M3DDPG | 43.75 ± 6.73 | 78.35 ± 3.27 | 72.73 ± 4.28 | 75.76 ± 4.43 |
| ROMANCE | 37.91 ± 7.12 | 85.63 ± 4.53 | 82.64 ± 5.83 | 80.37 ± 5.06 |
| Ours | 54.43 ± 6.36 | 89.85 ± 5.13 | 87.37 ± 3.63 | 84.46 ± 6.34 |

## D.5 EFFECT OF THE NUMBER OF STYLE CODES

**Research Question:** How does the number of style codes $K$ affect the performance of cooperative agents? Since style codes are used to capture teammate variability, it is important to understand whether increasing $K$ improves cooperation or introduces unnecessary complexity.

**Experiment Setup:** We vary the number of style codes $K \in \{2, 6, 10, 14\}$ and evaluate agents across four environments: Predator-Prey, 2s3z, 3m, and MMM. During training, the style code $z$ is uniformly sampled at the beginning of each episode and fixed throughout the rollout. The rest of the training setup is identical to the main experiments. Performance is measured by the episode return in Predator-Prey and win rates in the SMAC maps.

**Experiment Result:** As shown in Figure 7, the number of style codes significantly influences performance. In **Predator-Prey**, the performance is lowest when $K = 2$. Increasing $K$ to 6 and 10 yields substantial gains, while performance drops again at $K = 14$.

A similar trend is observed in the SMAC tasks. In **2s3z** and **3m**, performance steadily improves as $K$ increases from 2 to 10, but slightly decreases when $K = 14$. In **MMM**, $K = 10$ again achieves the highest win rate, while both smaller ($K = 2$) and larger ($K = 14$) values result in weaker performance.

These results suggest that a moderate number of style codes ($K = 6$-10) provides the best balance between representation capacity and training stability in most environments. However, we also observe that in relatively simple environments such as **Predator-Prey**, a smaller code number (e.g., $K = 6$) is sufficient or even preferable. This indicates that the optimal number of style codes is environment-dependent: more complex tasks benefit from richer style representations, while simpler tasks may be better served by a smaller and more efficient code number.

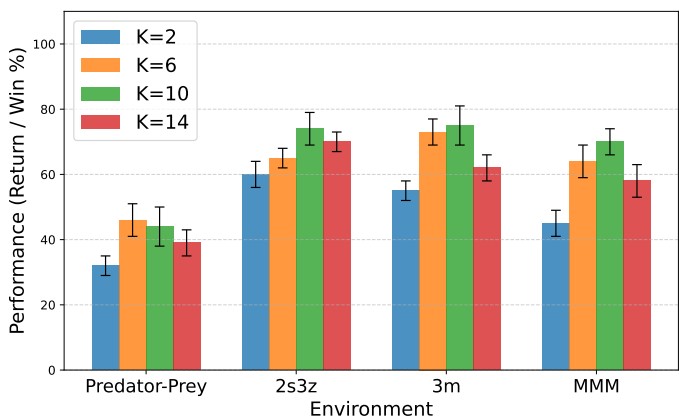

Figure 7: Performance comparison with different numbers of style codes $K$ across four environments, with error bars denoting 95% confidence intervals over 5 runs.

### D.6 ADDITIONAL ABLATION STUDY

Table 4 presents the ablation results evaluating the contribution of each component in our framework across both the Predator–Prey environment and three SMAC scenarios. The complete method ("Original") achieves the highest performance on all tasks, and removing any component leads to a noticeable degradation, demonstrating that each module plays a distinct and complementary role.

Removing CAPE results in a moderate decline across all environments, showing that adaptive adversarial context encoding is crucial for maintaining coordination under adversarial perturbations. Eliminating the multi-critic design also yields a large drop, especially in SMAC maps, highlighting the importance of decomposing heterogeneous value estimation signals during robust training.

The Instance-wise Relation Distillation (IRD) loss regularization also proves essential: without it, robustness drops substantially, indicating that continual alignment of adversarial embeddings mitigates catastrophic forgetting and stabilizes co-evolutionary learning. Removing components of the hierarchical encoder structure, either the local encoder or the global encoder, degrades performance across all four tasks, supporting our design choice that local trajectory features and global attention-based aggregation capture complementary information about adversarial behavior. Finally, removing the mutual-information objective in DAAG yields the most severe performance degradation, confirming that mutual-information-driven style diversification is a key factor in generating sufficiently rich adversarial populations for robust policy learning.

Overall, the ablations validate that robustness arises from the synergy of all components: diverse adversarial generation (MI), structured representation learning (CAPE + hierarchical encoders + IRD), and stable value estimation (multi-critic). Removing any part weakens this synergy and reduces the system's ability to generalize against strong and unseen adversaries.

## E IMPLEMENTATION DETAILS

### E.1 MODEL ARCHITECTURE

We now describe the network components used in our implementation. The **DAAG** module implements adversarial policy generation with multi-objective optimization, while the **CAPE** module is responsible for encoding agent trajectories into policy representations. Below, we summarize the sub-modules of each component, focusing on the network structures (e.g., Transformer blocks, MLPs, and LSTMs) that enable reproducibility.

**Diverse Adversarial Agent Generator (DAAG) / Multi-objective PPO.**

- **RL-Actor Net:** MLP projection followed by an LSTM (Hochreiter & Schmidhuber, 1997) for temporal feature extraction, with an action head producing the final action distribution.

Table 4: Ablation study results. **w/o CAPE** denotes removing the CAPE module from the original architecture. **w/o Multi-Critic** denotes replacing the Multi-Objective Critic in DAAG with a standard actor–critic structure. **w/o IRD**, **w/o Local Encoder**, and **w/o Global Encoder** respectively disable the continual distillation term, the local trajectory encoder, and the global attention encoder in CAPE. **w/o MI** removes the mutual-information objective in DAAG, preventing the generation of diverse adversarial styles. Best results in **bold**.

| Method | Pred–Prey ↑ | Δ | 2s3z (%) ↑ | Δ | 3m (%) ↑ | Δ | MMM (%) ↑ | Δ |
|---|---|---|---|---|---|---|---|---|
| Original | **46.23 ± 1.16** | — | **73.38 ± 2.42** | — | **75.46 ± 2.87** | — | **69.85 ± 3.17** | — |
| w/o CAPE | 41.25 ± 2.31 | −4.98 | 63.23 ± 2.33 | −10.15 | 58.74 ± 3.63 | −16.72 | 52.25 ± 3.28 | −17.60 |
| w/o Multi-Critic | 36.84 ± 2.76 | −9.39 | 68.79 ± 3.57 | −4.59 | 64.52 ± 4.83 | −10.94 | 57.24 ± 4.19 | −12.61 |
| w/o IRD | 38.42 ± 3.13 | −7.81 | 52.38 ± 3.23 | −21.00 | 47.82 ± 3.75 | −27.64 | 48.83 ± 4.26 | −21.02 |
| w/o Local Encoder | 35.15 ± 3.42 | −11.08 | 58.36 ± 4.34 | −15.02 | 61.48 ± 3.47 | −13.98 | 55.23 ± 4.87 | −14.62 |
| w/o Global Encoder | 43.79 ± 2.14 | −2.44 | 66.73 ± 3.86 | −6.65 | 65.82 ± 4.51 | −9.64 | 62.48 ± 3.73 | −7.37 |
| w/o MI | 32.58 ± 3.87 | −13.65 | 50.92 ± 2.92 | −22.46 | 57.25 ± 2.36 | −18.21 | 44.63 ± 4.64 | −25.22 |

- **RL-Critic Net:** Similar MLP + LSTM structure as the actor net.

- **Multi-objective Value Heads:** Multiple parallel MLP heads, each predicting a scalar for different objectives (e.g., task value, style value).

- **Style Predictor:** MLP classifier that predicts discrete style codes for disentanglement and auxiliary supervision.

- **Optimization:** PPO with clipped surrogate loss, GAE for advantage estimation, and entropy regularization.

**Contrastive Agent Policy Encoder (CAPE).**

- **Local Encoder:** The local encoder is implemented using stacked Transformer blocks with multi-head self-attention and feed-forward layers (Vaswani et al., 2017). Following the common practice in Transformer-based encoders (Dosovitskiy et al., 2020), we prepend a special $[CLS]$ token to each trajectory, and the final hidden state of this token is used as a compact representation of the entire trajectory.

- **Global Encoder:** The global encoder aggregates information across agents using attention pooling. Specifically, we introduce a learnable query token that attends to the set of local features (Lee et al., 2019), and the final hidden state of this token is taken as the global context representation $c$.

- **Positional Encoding:** Adds sinusoidal positional embeddings to trajectory sequences (Vaswani et al., 2017).

- **Projection Head:** MLP with LayerNorm and $\ell_2$ normalization, projecting features into a contrastive embedding space.

## E.2 TRAINING PROCEDURE

Our training procedure consists of two major phases: a *warm-up phase* and an *adversarial-cooperative alternating training phase*. The overall process is designed to progressively enhance both the Contrastive Agent Policy Encoder (CAPE) and the Diverse Adversarial Agent Generator (DAAG), while simultaneously improving the robustness and generalization ability of the cooperative agents, as shown in Algorithm 1.

**Warm-up Phase** At the beginning of training, we perform a warm-up stage without activating the diverse adversarial policy generation mechanism. In this stage, both the adversarial agent and the cooperative agents are trained independently with their respective objectives to acquire a basic level of competence:

- **Adversarial Agent:** Learns to disrupt the cooperative agents' performance using a standard PPO-based policy optimization.

- **Cooperative Agents:** Trained to maximize their task performance without facing diverse adversarial strategies.

---

**Algorithm 1** Co-evolutionary Robust MARL Training

---

**Predefine:** Number of cooperative agents $N$, style prior $p(z)$, DAAG replay buffer $\mathcal{B}$, CAPE memory buffer $\mathcal{M}$, alternation interval $T_{\text{alt}}$, total iterations $E_{\text{tot}}$
**Initialization:** Adversarial policy $\pi_\theta^{adv}$, style predictor $q_\omega$, CAPE encoders $f_\psi^{local}, f_\gamma^{global}$, MARL actor $\pi_\beta$ and critic $Q_\alpha$

 1: **Warm-up Phase:**
 2: Train adversarial agents $\pi_\theta^{adv}$ (w/o style conditioning)
 3: Train cooperative agents $\pi_\beta$ (w/o policy encoding)
 4: **Adversarial–Cooperative Co-Evolutionary Training Phase:**
 5: **for** iteration $= 1$ to $E_{\text{tot}}$ **do**
 6:     **Stage 1: DAAG Optimization**
 7:     **for** step $= 1$ to $T_{\text{alt}}$ **do**
 8:         Sample style code $z \sim p(z)$
 9:         Run $\pi_\theta^{adv}$ (Eq. 6) to collect trajectory $\tau^{adv}$
10:         Compute $r_t^{task}, r_t^{style}$ and store $(\tau^{adv}, z)$ in $\mathcal{B}$
11:         Update $q_\omega$ (Eq. 5)
12:         Update $\pi_\theta^{adv}$ (Eq. 9)
13:     **end for**
14:     **Stage 2: CAPE & Cooperative Policy Update**
15:     **for** step $= 1$ to $T_{\text{alt}}$ **do**
16:         Sample adversarial agents
17:         Collect $\{\tau_i\}_{i=1}^N$ from cooperative agents
18:         Encode $\tau_i \rightarrow c_t^i, c_t'$ (Eqs. 10, 12)
19:         Execute MARL policy with embedding (Eq. 18)
20:         Store $\{\tau_i\}$ into $\mathcal{M}$
21:         Update $f_\psi^{local}, f_\gamma^{global}$ (Eq. 17)
22:         Update $\pi_\beta, Q_\alpha$ (Eqs. 9, 19)
23:     **end for**
24: **end for**

---

This warm-up ensures that both sides have a reasonable initial policy foundation before entering the alternating training phase.

**Alternating Training with Diverse Adversarial Policy Generation** After the warm-up phase, we activate the Diverse Adversarial Agent Generator (DAAG). We sample different *style vectors $z$* from a predefined distribution to generate diverse adversarial policies. These style vectors modulate the adversarial agent's policy network, enabling the emergence of distinct behavioral patterns.

The training proceeds in alternating cycles:

1. **Adversarial Policy Training:** For $N_{\text{alt}}$ episodes, the adversarial agent is trained against the cooperative agents using a randomly selected style vector $z$ in each episode. This promotes policy diversity across different style vectors.

2. **Cooperative Policy and Policy Encoder Training:** After the adversarial update cycle, we switch to training the cooperative agents along with the CAPE. During this stage:

   - The cooperative agents face adversarial policies sampled from the pool of previously generated diverse policies.

   - The **policy encoder**, implemented with a Transformer-based architecture, processes long-horizon trajectories to extract high-level, discriminative representations of different adversarial behaviors.

   - The encoder learns to capture style-specific features that facilitate the cooperative agents to adapt to different types of adversarial policies.

### E.3 SUMMARY

This alternating process repeats for multiple cycles. Over time:

- The adversarial policies become increasingly diverse and effective in exploiting weaknesses of the cooperative policy.
- The cooperative policy learns to counter a broad range of adversarial policies by leveraging the policy encoder's discriminative embeddings.

Eventually, the cooperative agents acquire strong robustness and generalization capability against unseen adversarial agents.

### E.4 HYPERPARAMETERS

The hyperparameters used in our experiments are listed in Table 5.

Table 5: Hyperparameters for Predator-Prey and SMAC environments.

| Module | Hyperparameter | Predator-Prey | SMAC |
|---|---|---|---|
| **Reinforcement Learning** | $\varphi$: Discount Factor | 0.98 | 0.99 |
| | $\epsilon_{\text{clip}}$: PPO Clip Ratio | 0.1 | 0.1 |
| | $K$: PPO Update Frequency (Episode) | 30 | 20 |
| | $\alpha_{\text{actor}}$: Actor Learning Rate | $5 \times 10^{-4}$ | $5 \times 10^{-4}$ |
| | $\alpha_{\text{critic}}$: Critic Learning Rate | $5 \times 10^{-4}$ | $5 \times 10^{-4}$ |
| **Adversarial Training** | $N_{\text{step}}$: Total Training Steps | $1 \times 10^{7}$ | $1 \times 10^{7}$ |
| | $E_{\text{tot}}$: Alternating Iteration Number | 5 | 5 |
| **Diverse Adversarial Agent Generator** | $\lambda$style: Style Reward Weight | 0.5 | 0.5 |
| | $\lambda$task: Task Reward Weight | 1.0 | 1.0 |
| | $N_{\text{style}}$: Style Number | 6 | 10 |
| **Contrastive Agent Policy Encoder** | $\alpha_{\text{enc}}$: Encoder Learning Rate | $3 \times 10^{-4}$ | $3 \times 10^{-4}$ |
| | $U_{\text{enc}}$: Encoder Update Interval (Timesteps) | 500 | 2000 |
| | $B_{\text{enc}}$: Encoder Batch Size | 128 | 128 |
| | $d_{\text{loc}}$: Policy Local Dim | 6 | 15 |
| | $d_{\text{glb}}$: Policy Global Dim | 10 | 20 |
| | $\sigma_{\text{tmp}}$: Contrastive Temperature | 0.07 | 0.07 |
| | $\kappa_{\text{IRD}}$: IRD Loss Weight | 0.6 | 0.6 |

### E.5 ABBREVIATIONS

Table. 6 provides a summary of the abbreviations used in this paper.

Table 6: List of Abbreviations and Their Full Terms

| Abbreviation | Full Term |
|---|---|
| MARL | Multi-Agent Reinforcement Learning |
| MAS | Multi-Agent Systems |
| DAAG | Diverse Adversarial Agent Generator |
| CAPE | Contrastive Agent Policy Encoder |
| SMAC | StarCraft Multi-Agent Challenge |
| PP | Predator-Prey |
| VDN | Value Decomposition Network |
| COMA | Counterfactual Multi-Agent Policy Gradients |
| MADDPG | Multi-Agent Deep Deterministic Policy Gradient |
| PPO | Proximal Policy Optimization |
| MAPPO | Multi-Agent Proximal Policy Optimization |
| RRL | Robust Reinforcement Learning |
| M3DDPG | MiniMax Multi-agent Deep Deterministic Policy Gradient |
| SimCLR | Simple Framework for Contrastive Learning of Visual Representations |
| MoCo | Momentum Contrastive Learning |
| Dec-POMDP | Decentralized Partially Observable Markov Decision Process |
| CTDE | Centralized Training with Decentralized Execution |

## F CONCEPTS AND ABBREVIATIONS

In this section, we summarize the main concepts and abbreviations used throughout the paper.

### F.1 CONCEPTS

- **Robust MARL:** Multi-agent reinforcement learning methods that maintain stable performance when facing environment variations or adversarial attacks.

- **Cooperative Agents:** A group of agents that share a common goal and learn to collaborate to maximize the team reward.

- **Adversarial Agents:** Agents that intentionally act against the objectives of cooperative agents, aiming to minimize their performance or disrupt their policies.

- **Adversarial Training:** A training paradigm where agents learn policies that are robust against adversarial attacks varying from observation perturbation, environment disturbance, to adversarial actions.

- **Alternating Training:** A training scheme where adversarial agents and cooperative agents are trained in alternating phases to progressively improve both sides.

## G USE OF LARGE LANGUAGE MODELS

We used large language models (LLMs) only for language polishing and LaTeX formatting. All research ideas, experimental design, and analysis were solely conducted by the authors.

