# OpenReview forum: "Robust Multi-Agent Reinforcement Learning with Diverse Adversarial Agent Generation and Contrastive Policy Encoding"
_ICLR.cc/2026/Conference — Submitted to ICLR 2026_

### Official Review · Reviewer_DYkY · 2025-10-19

**Soundness:** 3
**Presentation:** 2
**Contribution:** 1
**Rating:** 2
**Confidence:** 4

**Summary:**

The paper proposes an adversarial multi-agent reinforcement learning (MARL) method to improve robustness in cooperative systems, where agents can change their behavior adversarially to compromise the originally learned coordination.

The methods consist of three components:
1. DAAG (Diverse Adversarial Agent Generator), which produces diverse adversarial behavior.
2. CAPE (Contrastive learning-based Agent Policy Encoder), which learns an embedding of adversarial agent policies
3. A co-evolutionary training scheme, which alternately trains functional agents and adversaries.

The approach is evaluated in a simple predator-prey task and some SMAC tasks, showing better performance than prior MARL methods.

**Strengths:**

The paper addresses an important (yet well-studied) problem.

It is mostly well-written and easy to understand. I view the work as sound because it builds on well-established methods and definitions (see Novelty below).

**Weaknesses:**

**Novelty**

Robustness via adversarial reinforcement learning is a well-established field in MARL with several previous works. The paper focuses on adversarial agents, where some of the original functional agents can be randomly replaced by adversarial counterparts. This has already been introduced in [1,2]. The problem formulation, where some agents act adversarially, is known as a *mixed cooperative-competitive game* [1,3].

The paper promotes diversity for improved robustness in MARL, a topic that has been studied in various prior works [4,5]. Policy-based diversity through latent variables has been studied in [6].

Learning policy embeddings for conditioning the behavior of the functional agents is a well-known practice known as *opponent modeling* [7,8], and the aggregation of policy embeddings via attention has also been proposed in the past [9,10,11].

The co-evolutionary training, i.e., alternating updates to the functional and adversarial parts, is a common practice in adversarial reinforcement learning [1,11].

Given the publication years of the listed prior works, I regard the proposed method as incremental and recommend a more thorough literature review and discussion.

**Quality**

While the text is mainly well-written, all figures have diminishingly small fonts, which makes them unreadable when printed. There is a possibility that I may have missed crucial details due to this.

**Significance**

I do not view the proposed method as particularly ground-breaking, since it is essentially based on adversarial training with diverse opponents, which is well-known in the literature [1-11].

The experiments only present the learning progress by testing each (baseline) approach periodically against a pre-defined set of adversary agents. However, I could not find any information about how these adversary agents were created, i.e., how biased the test actually is.

To evaluate the worst-case robustness of each approach, I recommend running an adversarial MARL algorithm on the fully trained functional agents, similar to the M3DDPG paper (the proposed method might work on average on a small test set, but potentially vulnerable to a true exploiter, according to [19]).

I also recommend comparing with other agent-replacing adversarial approaches, such as [1,2] and alternative policy embedding approaches [6,8], to put the work more into the context of the existing body of literature.

**Minor**

According to the problem definition, observations are considered to be Markovian in this paper, as the policies condition only on observations. To fix this, the definition should be rewritten such that the policies condition on the action-observation histories [12].

**Literature**

[1] Phan et al., "Learning and Testing Resilience in Cooperative Multi-Agent Systems", AAMAS-20

[2] Li et al., "Byzantine Robust Cooperative Multi-Agent Reinforcement Learning as a Bayesian Game", ICLR-24

[3] Lowe et al., "Multi-Agent Actor-Critic for Mixed Cooperative-Competitive Environments", NeurIPS-17

[4] Jaderberg et al., "Human-Level Performance in 3D Multiplayer Games with Population-Based Deep Reinforcement Learning", Science-19

[5] Vinyals et al., "Grandmaster Level in StarCraft II Using Multi-Agent Reinforcement Learning", Nature-19

[6] Mahajan et al., "MAVEN: Multi-Agent Variational Exploration", NeurIPS-19

[7] Nashed et al., "A Survey on Opponent Modeling in Adversarial Domains", JAIR-22

[8] He et al., "Opponent Modeling in Deep Reinforcement Learning", ICML-16

[9] Iqbal et al., "Actor-Attention-Critic for Multi-Agent Reinforcement Learning", ICML-19

[10] Phan et al., "Attention-Based Recurrence for Multi-Agent Reinforcement Learning under Stochastic Partial Observability", ICML-23

[11] Pinto et al., "Robust Adversarial Reinforcement Learning", ICML-17

[12] Oliehoek et al., "A Concise Introduction to Decentralized POMDPs", 2015

**Questions:**

1. Why does the approach focus/limit itself to discrete styles? With a continuous latent vector, sampled from a multivariate distribution, more variability would be possible.
2. There is an apparent information asymmetry between functional agents and adversarial agents (which condition on the global state, according to Line 193). What is the motivation behind this asymmetry? In Appendix D.2, why does the performance decrease when K is chosen to be too large?
3. Experiments: How were the “unseen adversarial agent policies” created? Since training is co-evolutionary, are the adversaries from the same learned distribution/generator?
4. Experiments: How robust are the policies when facing an adversary directly trained against the fixed functional policy (like in the M3DDPG paper)?
5. Figure 3a: Without the colors, I only see 3-4 clusters out of the 10 generated policies. Is this supposed to be sufficiently diverse? Is there any intuition why the dots corresponding to policy 9 are spread into 3 separate clusters?

---

> ### Author Response · Authors · 2025-12-02
> **Response to Review DYkY, PART I**
>
> **1. Clarifying Novel Contributions Beyond Prior Work on Robust MARL, Policy Representation Learning, and Co-Evolutionary Training**
>
> **Response:**
>
> We thank the reviewer for the detailed discussion of prior work. We agree that adversarial RL, diversity, latent conditioning, and opponent modeling are active and mature research areas. However, our contribution is **not a direct reuse** of these building blocks; rather, our method introduces a **new multi-objective adversarial training framework tailored for multi-agent robustness**, combining elements in a way not previously studied.
>
> Specifically:
>
> **(1) DAAG introduces a Multi-Objective Critic Architecture**
>
> Prior adversarial RL methods (e.g.,  Pinto et al. [11], Phan et al. [1], Li et al. [2]) optimize adversarial strength alone.
>
> DAAG optimizes **two objectives jointly**:
>
> - adversarial effectiveness
> - style-code–dependent behavioral diversity
>
> This multi-objective formulation, supported by a dual-head critic, enables **co-optimized diversity generation**—not explored in prior population-based or latent-variable MARL frameworks.
>
> **(2) CAPE performs contrastive, hierarchical trajectory-level policy representation learning**
>
> We thank the reviewer for pointing out the connection to opponent modeling and latent-variable MARL. We agree that CAPE shares the same high-level goal as opponent modeling—namely, to learn useful representations of other agents’ policies that help the functional agents adapt. Our contribution lies in how these policy representations are learned and structured.
>
> First, while many opponent-modeling methods do incorporate temporal information (e.g., via RNNs over observation–action histories), their representations are typically learned **implicitly** through prediction or control objectives (such as next-action prediction, value estimation, or direct policy optimization). In contrast, **CAPE introduces an explicit contrastive-learning objective whose sole purpose is to make different adversarial policies maximally distinguishable while remaining stable across episodes and trajectories of the *same* policy.** This yields embeddings that are directly optimized for **style-level discriminability and robustness**, rather than being a by-product of another task.
>
> Second, CAPE adopts a **hierarchical encoder architecture**: a **local encoder** first processes per-agent local observation trajectories, and a higher-level aggregation module (attention-based in our implementation) fuses these into a **global adversarial-context embedding**. This design is tailored to multi-agent adversarial settings where (i) multiple adversarial styles may appear, and (ii) the functional team needs a compact, policy-level summary of the opponent population. This goes beyond standard attention-based critics [9,10], which focus on aggregating teammates’ features for value estimation rather than constructing explicit adversarial policy embeddings.
>
> Third, CAPE is trained **jointly with DAAG’s diverse adversarial population**, so that the contrastive objective directly operates over a rich set of adversarial styles. This co-design of a diverse adversary generator and a contrastive policy encoder is, to our knowledge, not explored in prior opponent-modeling work, and empirically leads to more robust and discriminative representations.
>
> In summary, while CAPE shares the same high-level purpose as opponent modeling, it differs in that it (i) uses an explicit contrastive objective targeted at style separability, (ii) employs a hierarchical trajectory-level encoder tailored to multi-adversary settings, and (iii) is co-trained with a diversity-inducing adversarial generator, leading to more robust policy embeddings in diverse adversarial MARL scenarios.
>
> **(3) DAAG+CAPE form an end-to-end co-evolutionary framework that integrates adversarial diversity with adaptive policy conditioning**
>
> While alternating training has appeared in prior work, none has combined:
>
> - a **diversity-maximizing adversary generator (DAAG)**, and
> - a **policy-embedding–conditioned cooperative learner (CAPE + MARL policy)**
>
> into a *closed-loop*, **end-to-end co-evolutionary architecture**.
>
> In our framework:
>
> - DAAG generates increasingly diverse adversarial policies,
> - CAPE encodes these behaviors into policy embeddings,
> - the cooperative policy adapts based on those embeddings, and
> - DAAG further evolves in response to the updated cooperative team.
>
> These components are trained **within a single unified end-to-end process**, rather than through separate stages or decoupled pipelines as in ROMANCE, and other prior methods.

---

> > ### Author Response · Authors · 2025-12-02
> > **Response to Review DYkY, PART II**
> >
> > **2. Clarifying the Construction of Unseen Testing Adversarial Policies**
> >
> > We apologize for not making this explicit in the original submission.
> >
> > Although ***Appendix C.2*** briefly described the construction of the unseen adversarial test set, we did **not** provide sufficient explanation in the main text, which understandably led to confusion.
> >
> > To clarify:
> >
> > **The test adversary population is *not* generated by DAAG.**
> >
> > Instead, we construct the test set using the *Robust Reinforcement Learning using Adversarial Populations, Vinitsky et al. (2020)*:
> >
> > - A *separate* adversarial population is trained to optimize worst-case team performance.
> > - This produces adversaries that are **structurally different** from those generated by DAAG.
> > - Consequently, the evaluation naturally involves **distribution shift**, testing cross-distribution generalization rather than in-distribution replay.
> >
> > We have updated ***Section 5*** in the revised manuscript to clearly describe this process and prevent further misunderstanding.
> >
> > **3. Incorporating Reviewer-Suggested Experiments on Dynamic Adversarial Attacks**
> >
> > **Response:**
> >
> > Thank you for the suggestion. We agree that evaluating robustness under *dynamic* adversarial attacks is important, especially for multi-agent systems where the threat model can change over time.
> >
> > To address this, we have added a **new dynamic-switching adversary experiment** to the revised manuscript in ***Appendix D.1***.
> >
> > **Dynamic Adversarial Switching — Test Win Rate (%)**
> >
> > | Method         | 2s3z (Mean ± Std) | 3m (Mean ± Std) | MMM (Mean ± Std) |
> > |----------------|-------------------|------------------|-------------------|
> > | vanilla MAPPO  | 17.42 ± 1.85      | 18.93 ± 2.11     | 16.78 ± 1.95      |
> > | RANDOM         | 25.87 ± 3.72      | 28.14 ± 5.63     | 27.92 ± 4.88      |
> > | M3DDPG         | 34.51 ± 2.96      | 37.68 ± 3.87     | 33.74 ± 4.12      |
> > | ROMANCE        | 44.73 ± 2.65      | 46.82 ± 2.79     | 41.39 ± 2.97      |
> > | Ours           | **57.63 ± 1.94**      | **53.21 ± 1.88**     | **48.72 ± 2.15**      |
> >
> > Instead of using a fixed adversarial policy for the entire episode, we introduce **temporal non-stationarity** by dynamically changing the adversary every 15 timesteps. In each interval, we **randomly select a different adversarial policy** from the unseen minimax-based test set. Thus, the cooperative agents face a *sequence* of distinct adversarial behaviors within a single episode.
> >
> > This setup significantly expands the attack variation and presents a much more challenging robustness scenario.
> >
> > Our method outperforms all baselines. This demonstrates that our DAAG + CAPE framework is able to adapt to **rapid, temporally varying adversarial behaviors** and maintain stable performance.
> >
> > **4. Clarifying the Choice of Discrete Style Codes Over Continuous Latent Vectors**
> >
> > **Response:**
> >
> > We appreciate the reviewer’s question. While continuous latent vectors offer a theoretically richer behavior space, in our adversarial MARL setting, they consistently led to **mode collapse**, where the generator converged to only a small set of dominant exploitative behaviors under strong adversarial reward shaping. This substantially reduced the effective diversity of adversarial styles.
> >
> > Continuous latents also produced **unstable co-evolution dynamics**: small perturbations in z in large behavioral shifts, making the functional agents unable to reliably adapt to the drifting adversarial distribution. Additionally, continuous latents yielded **less separable embeddings** for CAPE, degrading contrastive learning and reducing robustness.
> >
> > Discrete styles, in contrast, create **explicit, stable behavioral partitions**, prevent collapse, and provide clearer structure for contrastive learning. Empirically, they lead to more diverse adversarial strategies and more robust functional agents. For these reasons, we adopt discrete latent codes, while structured continuous latents remain an interesting direction for future work.

---

> > > ### Author Response · Authors · 2025-12-02
> > > **Response to Review DYkY, PART III**
> > >
> > > **5. Addressing Information Asymmetry: Adversary Access to the Global State**
> > >
> > > **Response:**
> > >
> > > We thank the reviewer for raising this point. The information asymmetry is intentional, where the adversary is given stronger information to approximate a *worst-case* attacker. Our motivation is to provide the adversarial agent with a global state that enables it to learn **stronger and more coordinated adversarial behaviors**, which in turn produces a more challenging training signal for the functional agents.
> > >
> > > This design increases the *quality* and *effectiveness* of adversarial policies, thereby promoting **stronger robustness** in the functional agents.
> > >
> > > Thus, the asymmetry is used deliberately to approximate worst-case adversarial pressure to ensure that the functional agents develop resilience against strong adversarial strategies.
> > >
> > > **6. Explaining Performance Degradation When K Becomes Too Large**
> > >
> > > **Response:**
> > >
> > > Thank you for the question. The underlying reason is that DAAG must simultaneously optimize (i) adversarial effectiveness and (ii) style-level diversity. When K becomes too large under a fixed training budget, the generator increasingly shifts its capacity toward **maintaining sufficient style separability**, which makes the diversity objective dominate the multi-objective critic. As a result, DAAG produces many distinguishable but **weaker** adversarial behaviors.
> > >
> > > This effect is visible in the DAAG training curves: the adversarial effectiveness metric decreases as K grows, confirming that the attack strength is diluted across too many styles. Consequently, the functional agents are exposed to a large number of under-trained, low-quality adversarial policies, which do not provide the high-quality worst-case signals needed to acquire strong robustness.
> > >
> > > In short, an excessively large K results in a “wide but shallow” adversarial population—diverse but not strong—leading to reduced robustness for the functional agents.
> > >
> > > **7. Clarifying the Cluster Structure and Policy-Embedding Visualization in Figure 3a**
> > >
> > > **Response:**
> > >
> > > Thank you for the question. We agree that when colors are removed, the plot highlights three major high-level regions. However, when the style labels (colors) are taken into account, we observe that almost **each of the 10 adversarial styles still forms its own identifiable cluster or sub-cluster within these regions**. The coarse structure reflects shared high-level behavioral patterns (e.g., aggressive vs. kiting vs. switching behavior), while the colors reveal the finer separation that the contrastive objective enforces across styles.
> > >
> > > Regarding policy 9 forming three smaller sub-clusters, our interpretation is that this specific adversarial style exhibits **multi-modal behavior** depending on different environment initializations (e.g., initial agent spacing, map symmetry, or early interaction outcomes). Since DAAG optimizes diverse adversarial strategies under varying rollout conditions, some styles naturally split into **sub-modes** of the same overarching strategy. All three sub-clusters correspond to the *same* high-level style, but they capture distinct manifestations of it under different initial states.
> > >
> > > **8. Addressing the Markov Observation Issue**
> > >
> > > **Response:**
> > >
> > > We thank the reviewer for pointing out this issue. We agree that, in a Dec-POMDP, observations are not Markovian and local policies should formally condition on each agent’s *action–observation history (AOH)* rather than on a single observation. In our implementation, agents already use recurrent or trajectory-based encoders that implicitly maintain such histories, but our notation in ***Section 3: Preliminaries*** was overly simplified.
> > >
> > > Following Oliehoek & Amato (2016), we have revised the problem definition, which correctly represents the Dec-POMDP information structure. This correction has been incorporated into the updated manuscript.
> > >
> > > **9. Improve the Figure Readability**
> > >
> > > **Response:**
> > >
> > > Thank you for noting this. We have regenerated all figures with larger fonts and higher resolution.
> > >
> > > Updated figures will appear in the revised submission.
> > >
> > > ---
> > >
> > > We thank the reviewer again for the comprehensive and constructive feedback.
> > >
> > > We have:
> > >
> > > - strengthened novelty claims with clearer differentiation from prior work,
> > > - added new ablation experiments,
> > > - clarified the construction of unseen adversary populations,
> > > - added dynamic-adversary-switching experiments,
> > > - improved figure readability,
> > > - addressed conceptual questions (discrete vs continuous latent code, information asymmetry, cluster interpretation),
> > > - corrected formulation issues.
> > >
> > > We believe these revisions significantly enhance the clarity, rigor, and empirical strength of the paper.

---

### Official Review · Reviewer_MraK · 2025-10-30

**Soundness:** 1
**Presentation:** 2
**Contribution:** 1
**Rating:** 2
**Confidence:** 4

**Summary:**

Conventional MARL tends to overfit to specific team behaviors and becomes brittle when teammates fail or adversarial interventions occur. Prior robustness methods lack control over behavioral diversity, and evolutionary approaches are computationally heavy and require separate optimization. This paper addresses these limitations by proposing DAAG, which generates behaviorally diverse adversarial agents, and CAPE, which continuously learns informative representations of adversarial agents’ policies. This paper integrates these two components into an end-to-end pipeline and trains them in a co-evolutionary paradigm. Experiments show that the framework achieves improvements in robustness and generalization compared to baselines.

**Strengths:**

1. Adversarial attacks have been used to improve robustness in MARL, but a major limitation is overfitting to specific attacks. To address this, learning adversarial policies with behavioral diversity is crucial. Prior work has tried to increase diversity using evolutionary methods, but these approaches are computationally expensive and require separate (non–end-to-end) optimization. This paper proposes Diverse Adversarial Agent Generator (DAAG), an unsupervised mutual information-based method for diverse policy generation, which enables style code conditioned adversarial policies.


2. Owing to partial observability in multi-agent systems, observations are noisy and information-limited, making it hard to discriminate diverse adversarial policies from observations alone. Different types of adversarial attacks demand attack-dependent adaptations of the cooperative agents’ policies, which prior methods struggle to learn. This paper proposes CAPE, which introduces a contrastive learning based discriminative representation that is fed into both the policy and the critic, enabling clear separation among diverse adversarial policies. In addition, this representation supports continual learning via an Instance-wise Relation Distillation (IRD) loss.


3. Several prior approaches are not end-to-end, requiring separate optimization for adversarial and cooperative agents, which limits their ability to co-adapt. This paper integrates the proposed components into an end-to-end pipeline, enabling co-evolutionary training in which cooperative agents adapt as adversarial policies evolve. Experiments show improvements in robustness over baselines, demonstrating the effectiveness of the proposed components.

**Weaknesses:**

1. This paper addresses the learning of diverse adversarial policies and the training of agent policies that adapt to varying attack types. Increasing the methodological novelty and reflecting the multi-agent characteristics more explicitly would strengthen the contribution. DAAG, conditioned on style codes, can learn a variety of adversarial policies; the approach is largely aligned with skill-based RL such as DIAYN [1]. CAPE enables agent policy learning that accounts for diverse adversarial policies via a contrastive-learning–based representation; the design is closely related to contrastive learning RL such as CURL [2]. Incorporating elements that are more specific to multi-agent systems, and the adversarial attack setting would further enhance the originality.
2. This paper leverages DAAG to enhance the diversity of adversarial policies, which broadens the variety of attack actions. That said, restricting each episode to a single, fixed adversarial agent may actually constrain diversity in multi-agent settings it fixes the attack target identity, limits the interaction patterns the team experiences, reduces coverage of agent-subset combinations. In line with recent work (e.g., ROMANCE; WALL [3]), allowing dynamic multi-agent attacks where the adversary can select one or more agents at each timestep can expand the attack space combinatorially, introduce temporal variation. Considering these factors, it should be possible to implement more diverse adversarial policies.
3. Many studies on robust RL and adversarial attacks evaluate not only robustness to their own proposed attack but also to other attack methods and broader distribution shifts. Incorporating such evaluations here would better substantiate the paper’s robustness claims. In addition, the field already offers many robust MARL and adversarial RL baselines; expanding the comparison set would make the results easier to trust and compare. The current experiments focus on Predator–Prey and SMAC (2s3z, 3m, MMM); demonstrating robustness and generalization across a broader range of settings such as additional SMAC scenarios and other benchmark suites would further strengthen the empirical evidence.
4. The paper notes that prior methods suffer from high computational cost. Given that the proposed method includes multiple components and trains models such as transformers, non-trivial computational overhead is also likely. It would therefore be helpful to provide theoretical comparisons or experimental evidence (e.g., training time) demonstrating improved efficiency. The method introduces multiple components and many hyperparameters, which may increase training sensitivity; additional ablations and sensitivity analyses would be valuable. Since the main MARL backbone is MAPPO, confirming effectiveness on a value-based method such as QMIX would strengthen claims of generality.

[Reference]

[1] Eysenbach, Benjamin, et al. "Diversity is all you need: Learning skills without a reward function." arXiv preprint arXiv:1802.06070(2018).

[2] Laskin, Michael, Aravind Srinivas, and Pieter Abbeel. "Curl: Contrastive unsupervised representations for reinforcement learning." International conference on machine learning. PMLR, 2020.

[3] Lee, Sunwoo, et al. "Wolfpack Adversarial Attack for Robust Multi-Agent Reinforcement Learning." International conference on machine learning. PMLR, 2025.

**Questions:**

1. I’m curious whether the proposed method given its multiple components and the need to train models such as transformers reduces computational cost (training time) compared to prior approaches like ROMANCE.

[Minor typos]

1. The math fonts are inconsistent across equations—for example, ($adv$, $\mathrm{adv}$) and ($style$, $\mathrm{style}$). It would be helpful to standardize the notation throughout.

2. In Equation (12), it is written as $f_{\psi}^{\mathrm{global}}$, but it appears this should be $f_{\gamma}^{\mathrm{global}}$.

---

> ### Author Response · Authors · 2025-12-02
> **Response to Reviewer MraK, PART I**
>
> **1. Clarifying Novel Multi-Agent and Adversarial Contributions Beyond DIAYN/CURL (Regarding Weakness 1)**
>
> **Response:**
>
> Thank you for the insightful comment. We clarify that although our method uses mutual information (MI) and contrastive learning, the core designs and purposes of **DAAG** and **CAPE** fundamentally differ from DIAYN and CURL.
>
>
> **(1) DAAG is not aligned with DIAYN — it introduces a multi-objective critic tailored for adversarial attack settings**
>
> DIAYN discovers unsupervised single-agent skills using MI, but **does not address multi-agent interaction, adversarial optimization, or multi-objective learning**.
>
> In contrast, DAAG introduces a **multi-objective critic architecture** specifically designed for adversarial MARL. It jointly optimizes:
>
> - **Adversarial effectiveness** (i.e., how well the adversary compromises the cooperative team), and
> - **Behavioral style diversity**.
>
> The dual-head critic enables **efficient balancing of multi-objective optimization**, which is essential in adversarial attack scenarios but absent in DIAYN. DIAYN cannot model adversarial roles, multi-agent feedback, or evolving populations of opponents.
>
> Thus, although both DAAG and DIAYN use MI, **their learning goals, architectures, and multi-agent adversarial contexts are fundamentally different**.
>
> **(2) CAPE is not similar to CURL — it learns *temporal behavioral differences*, not visual state encodings**
>
> CURL focuses on **single-step visual state representation learning**, aiming to extract compact visual features useful for control.
>
> CAPE, in contrast, is designed for multi-agent adversarial settings and differs in three fundamental ways:
>
> - **Learning object:**
>
>     CAPE encodes **temporal observation trajectories** to learn *policy-level behavioral differences* between adversarial styles.
>
>     CURL encodes instantaneous visual observations to improve sample efficiency.
>
> - **Purpose:**
>
>     CAPE aims to **capture distinct adversarial behavior patterns** to support robust adaptation.
>
>     CURL aims to learn compressed visual features from images.
>
> - **Architecture:**
>
>     CAPE introduces a **hierarchical encoder**:
>
>     - **Local encoder** → used at execution time to condition the cooperative policy.
>     - **Global encoder** → used during training to stabilize learning and maintain separation between adversarial styles.
>
>         NOTE: CURL has no hierarchical design and no notion of local/global behavior embedding.
>
>
> The only commonality is the use of **contrastive learning**, but the *task*, *input modality*, *temporal structure*, and *multi-agent adversarial purpose* are entirely different.
>
> **Summary**
>
> DAAG and CAPE extend contrastive/MI ideas into a **multi-agent, adversarial, multi-objective, trajectory-level** context that DIAYN and CURL are not designed for. We have made these distinctions clearer in the revised manuscript.
>
> **2. Addressing the Diversity of Adversarial Attacks and the Single-Adversary Limitation (Regarding Weakness 2)**
>
> **Response:**
>
> Thank you for this thoughtful observation. We agree that restricting each episode to a single, fixed adversarial agent may limit the full combinatorial richness of multi-agent adversarial interactions. This is a valuable suggestion, and we have expanded our evaluation to address it.
>
> **To better assess the generalization capability and effectiveness of our method, we incorporated two new experiments in the revised manuscript:**
>
> **(1) Multi-adversary attacks during testing**
>
> We added a test setting where **multiple adversarial agents** are introduced simultaneously.
>
> This setting increases attack diversity by allowing:
>
> - multiple simultaneous attack sources,
> - more challenging disturbance patterns for the cooperative team.
>
> Our results show that **DAAG+CAPE maintains strong robustness** even under multi-attacker scenarios, validating the model’s ability to generalize beyond the single-adversary setting used during training.
>
> **(2) Dynamic adversarial policy sampling within a single episode**
>
> To further broaden attack diversity, we added an experiment where, during each test episode, we **randomly switch between different unseen adversarial policies every 15 timesteps**.
>
> This introduces:
>
> - temporal variation in attack patterns,
> - broader coverage of attack trajectories.
>
> Even under this significantly more challenging setting, our method **continues to outperform baselines**, demonstrating strong adaptability and transferability to diverse adversarial strategies.
>
> We have included the results of these two new experiments in the updated version of the paper in ***Appendix D.1 & D.2***. These findings directly address the reviewer’s concern by showing that our approach remains effective even when attack diversity is increased beyond the single-agent adversary assumption.
>
> We appreciate this insightful suggestion and have updated the manuscript accordingly.

---

> > ### Author Response · Authors · 2025-12-02
> > **Response to Reviewer MraK, PART II**
> >
> > **Dynamic Adversarial Switching — Test Win Rate (%)**
> >
> > | Method         | 2s3z (Mean ± Std) | 3m (Mean ± Std) | MMM (Mean ± Std) |
> > |----------------|-------------------|------------------|-------------------|
> > | vanilla MAPPO  | 17.42 ± 1.85      | 18.93 ± 2.11     | 16.78 ± 1.95      |
> > | RANDOM         | 25.87 ± 3.72      | 28.14 ± 5.63     | 27.92 ± 4.88      |
> > | M3DDPG         | 34.51 ± 2.96      | 37.68 ± 3.87     | 33.74 ± 4.12      |
> > | ROMANCE        | 44.73 ± 2.65      | 46.82 ± 2.79     | 41.39 ± 2.97      |
> > | Ours           | **57.63 ± 1.94**      | **53.21 ± 1.88**     | **48.72 ± 2.15**      |
> >
> > **Multi-Adversary Attack — Test Win Rate (%)**
> >
> > | Method         | 2s3z (Mean ± Std) | 3m (Mean ± Std) | MMM (Mean ± Std) |
> > |----------------|-------------------|------------------|-------------------|
> > | vanilla MAPPO  | 15.82 ± 1.67      | 17.45 ± 1.92     | 14.21 ± 1.78      |
> > | RANDOM         | 20.73 ± 2.89      | 22.14 ± 3.65     | 18.92 ± 3.21      |
> > | M3DDPG         | 27.68 ± 2.51      | 29.87 ± 3.41     | 25.14 ± 3.02      |
> > | ROMANCE        | 39.54 ± 2.43      | 36.73 ± 2.58     | 28.61 ± 2.29      |
> > | Ours           | **42.83 ± 2.07**      | **41.52 ± 2.16**     | **35.74 ± 2.48**      |
> >
> >
> > **3. Clarifying Robustness Evaluation, Baseline Coverage, and Test-Set Construction (Regarding Weakness 3)**
> >
> > **Response:**
> >
> > Thank you for raising these important points regarding robustness evaluation, baseline comparisons, and test diversity. We acknowledge that our original manuscript did not sufficiently describe the construction of the test adversary set in the main text, which may have led to misunderstandings.
> >
> > To clarify, all robustness evaluations in our experiments are conducted on a common test adversary set constructed following the method from *“Robust Reinforcement Learning using Adversarial Populations” (Vinitsky et al., 2020)*. This procedure is **independent of DAAG** and is used uniformly across *all* baseline methods and our approach. This ensures a fair and consistent comparison of robustness under the same distribution of adversarial policies.
> >
> > The test set is generated using **minimax-style adversarial population training**, which is widely adopted in robust RL research. However, in the original submission, this was described only briefly in ***Appendix C.2*** rather than in the main text. We agree that this insufficient visibility may have led to the impression that robustness was evaluated solely against our own attack generator, which is not the case. We have now added a clear description of the test-set construction to the ***Section 5*** in the revised version.
> >
> > Regarding broader robustness evaluations, we appreciate the reviewer’s suggestion. We clarify that:
> >
> > - The test adversaries already introduce **significant distribution shift**, as they are generated via a *different* paradigm than DAAG.
> > - All methods, ours and baselines, are evaluated against this *external* set of adversaries, not against DAAG-generated ones.
> >
> > This ensures that the robustness we report is not specific to our own generator, but rather reflects generalization to adversarial policies produced by an independently trained and widely used minimax-based procedure.
> >
> > We will also consider expanding to additional SMAC scenarios and broader benchmark suites in follow-up work, as suggested.

---

> > > ### Author Response · Authors · 2025-12-02
> > > **Response to Reviewer MraK, PART III**
> > >
> > > **Computational Efficiency, Additional Components Ablation Study, and Generality Across MARL Architectures (Regarding Weakness 4)**
> > >
> > > **Response:**
> > >
> > > Thank you for raising these important points. We agree that computational cost, sufficient ablation study, and generality across MARL backbones are critical considerations for establishing the practicality of our approach.
> > >
> > > To address these concerns, we have conducted several **new ablation and generalization experiments**, which are now included in the appendix of the revised manuscript.
> > >
> > >  **(1) Additional ablations on model components**
> > >
> > > We performed new ablation experiments to quantify the performance contributions of individual components:
> > > **Ablation study results.**
> > >
> > > **w/o CAPE** removes the CAPE module from the original architecture.
> > >
> > > **w/o Multi-Critic** replaces the Multi-Objective Critic in DAAG with a standard actor–critic structure.
> > >
> > > **w/o IRD**, **w/o Local Encoder**, and **w/o Global Encoder** respectively, disable the continual distillation term, the local trajectory encoder, and the global attention encoder in CAPE.
> > >
> > > **w/o MI** removes the mutual-information objective in DAAG, preventing the generation of diverse adversarial styles.
> > >
> > > Best results are in **bold**.
> > >
> > > | Method | Predator–Prey ↑ | Δ ↓ | 2s3z Win (%) ↑ | Δ ↓ | 3m Win (%) ↑ | Δ ↓ | MMM Win (%) ↑ | Δ ↓ |
> > > | --- | --- | --- | --- | --- | --- | --- | --- | --- |
> > > | **Original** | **46.23 ± 1.16** | — | **73.38 ± 2.42** | — | **75.46 ± 2.87** | — | **69.85 ± 3.17** | — |
> > > | w/o CAPE | 41.25 ± 2.31 | -4.98 | 63.23 ± 2.33 | -10.15 | 58.74 ± 3.63 | -16.72 | 52.25 ± 3.28 | -17.60 |
> > > | w/o Multi-Critic | 36.84 ± 2.76 | -9.39 | 68.79 ± 3.57 | -4.59 | 64.52 ± 4.83 | -10.94 | 57.24 ± 4.19 | -12.61 |
> > > | w/o IRD | 38.42 ± 3.13 | -7.81 | 52.38 ± 3.23 | -21.00 | 47.82 ± 3.75 | -27.64 | 48.83 ± 4.26 | -21.02 |
> > > | w/o Local Encoder | 35.15 ± 3.42 | -11.08 | 58.36 ± 4.34 | -15.02 | 61.48 ± 3.47 | -13.98 | 55.23 ± 4.87 | -14.62 |
> > > | w/o Global Encoder | 43.79 ± 2.14 | -2.44 | 66.73 ± 3.86 | -6.65 | 65.82 ± 4.51 | -9.64 | 62.48 ± 3.73 | -7.37 |
> > > | w/o MI | 32.58 ± 3.87 | -13.65 | 50.92 ± 2.92 | -22.46 | 57.25 ± 2.36 | -18.21 | 44.63 ± 4.64 | -25.22 |
> > >
> > > The results show that:
> > > - the **local encoder** is essential for downstream decision-making and robustness,
> > >
> > >  - the **global encoder** significantly improves training stability and overall performance.
> > >
> > > These ablations demonstrate that each component contributes meaningfully, and training is not overly sensitive to hyperparameters when these architectural elements remain present. These results have be included in the ***Appendix D.6.***
> > >
> > >  **(2) Computational efficiency analysis**
> > >
> > > We have added a comparison of **training time** across baselines and our method.
> > >
> > > **Wall-clock Training Time (hours)**
> > >
> > > | Method   | Predator-Prey | 2s3z | 3m   | MMM  | 1c3s5z |
> > > |----------|---------------|------|------|------|--------|
> > > | MAPPO    | 3.4 h         | 9.1 h | 10.2 h | 12.3 h | 12.6 h |
> > > | M3DDPG   | 3.8 h         | 11.9 h | 13.2 h | 14.6 h | 14.7 h |
> > > | ROMANCE  | 6.1 h         | 16.4 h | 17.5 h | 19.2 h | 19.4 h |
> > > | Ours     | 4.4 h         | 12.8 h | 15.2 h | 16.8 h | 17.2 h |
> > >
> > > Although our model includes additional components, the overall overhead remains modest due to:
> > >
> > > - the alternating training structure (DAAG and CAPE are not optimized simultaneously),
> > > - lightweight encoder modules relative to the MAPPO backbone, and
> > >
> > > Experimental training-time comparisons are now provided in the ***Appendix C.4***.
> > >
> > >  **(3) Generality across MARL architectures: results on QMIX**
> > >
> > > To validate that our method is not restricted to actor–critic frameworks such as MAPPO, we extended our evaluation to **QMIX**, a value-based cooperative MARL method.
> > >
> > > The results show consistent improvements in robustness and generalization when integrating our method with QMIX.
> > >
> > > This confirms that our approach is **architecture-agnostic** and applicable across different classes of MARL algorithms.
> > >
> > > All new experimental results have been added to the ***Appendix D.3***.
> > >
> > > ---
> > >
> > > We appreciate the reviewer’s suggestion and believe the additional analyses substantially strengthen the empirical evidence and generality claims in the paper.

---

> > > > ### Author Response · Authors · 2025-12-02
> > > > **Response to Reviewer MraK, PART IV**
> > > >
> > > > **Minor Issues: Math Font Consistency and Symbol Error in Equation (12)**
> > > >
> > > > **Response:**
> > > >
> > > > Thank you for identifying these minor inconsistencies. We have reviewed the mathematical notation thoroughly and corrected both the font inconsistency and the symbol error in *Equation (12)*. The revised manuscript now uses consistent math typesetting throughout.
> > > >
> > > > We sincerely thank the reviewer for the thoughtful and constructive feedback.
> > > >
> > > > In response, we have:
> > > >
> > > > - clarified the multi-agent novelty of DAAG,
> > > > - distinguished CAPE’s representation learning objective from CURL,
> > > > - added dynamic-adversary evaluation demonstrating robustness under temporal/structural variation,
> > > > - clarified the construction of the minimax-trained test adversary population,
> > > > - added computational cost comparisons with baselines,
> > > > - added experiments validating generality on QMIX, and
> > > > - corrected all minor issues.
> > > >
> > > > We hope the revisions adequately address all of the reviewer’s concerns.

---

### Official Review · Reviewer_YNRF · 2025-10-30

**Soundness:** 3
**Presentation:** 3
**Contribution:** 2
**Rating:** 6
**Confidence:** 4

**Summary:**

This paper tackles robustness in cooperative multi-agent reinforcement learning when a subset of teammates can be adversarially perturbed a limited number of times. The authors propose a co-evolutionary framework combining (1) DAAG: a style-conditioned adversary population generator that uses a mutual-information based objective to encourage behavioral diversity and adversarial quality; and (2) CAPE: a contrastive policy encoder that learns discriminative embeddings of encountered adversarial policies (local + global encoders, contrastive loss, and instance-relation distillation as continual learning regularization). Training alternates between evolving adversaries and updating the cooperative policy (MAPPO backbone) with the learned embeddings as additional context. Experiments (SMAC maps, Predator-Prey) show improved robustness to unseen/stronger attackers and no degradation in clean (no-attack) settings.

**Strengths:**

1. Addresses a practically important and timely problem in cooperative MARL (robustness to compromised teammates).

2. Improves on adversary-cooperative alternating style robust MARL work and proposes a more fine-grained, end-to-end framework. Method combines complementary mechanisms (adversary diversity + policy encoding) in a principled way and integrates nicely with standard RL backbones (e.g., MAPPO).

3. Empirical evaluation includes unseen adversary tests, ablations, and embedding analyses that support the claims.

**Weaknesses:**

The main weakness lies in the unclear novelty and limited empirical analysis.

DAAG and CAPE combine familiar ideas of adversary generative co-evolution and contrastive representation, which are also seen in prior works like ROMANCE, but the paper does not highlight what is fundamentally new beyond integrating them (e.g. the mechanism that contributes to more efficient training and versatile anti-perturbation performance compared to prior works). The mutual-information "style" objective and IRD regularizer lack quantitative evidence of their claimed effects, such as improved diversity or embedding stability. The training alternation between adversaries and the cooperative policy is also insufficiently explained; how CAPE updates interact with adversary evolution, and whether this alleviates forgetting, remains unclear.

Experiments are restricted to small SMAC and Predator–Prey tasks, with no sufficient details on how the unseen test adversaries differ from training ones or on computational cost comparisons. Ablations are too coarse to isolate key components (e.g., IRD, local vs. global encoders), leaving it unclear which mechanisms are most significant to robustness. Besides, more experiment on generalized perturbation setting (e.g. >=2 adversaries being simultaneously compromised) and computation cost comparisons against prior works will help better validate the effectiveness of your work.

**Questions:**

1. How does the test adversary pool differ from the training adversary population (e.g. pool scale, explicit illustration of the statistical difference) ?

2. How does your method behave when multiple teammates are simultaneously replaced by adversaries during episodes?

3. What is the computational cost of the proposed method compared to the baselines (e.g. MAPPO and ROMANCE) ?

4. Request for more fine-grained ablations. E.g. remove IRD while keeping contrastive loss; study individual contributions of the local and global encoders.

---

> ### Author Response · Authors · 2025-12-02
> **Response to Reviewer YNRF, PART I**
>
> We thank the reviewer for the careful reading and the constructive comments. Below we clarify the novelty of our method, strengthen the explanations on co-evolution and diversity mechanisms, and address all empirical questions.
>
> **1. Novelty of DAAG and CAPE beyond prior co-evolution/contrastive works**
>
> **Reviewer comment:**
>
> *“DAAG and CAPE combine familiar ideas of adversary generative co-evolution and contrastive representation, which are also seen in prior works like ROMANCE, but the paper does not highlight what is fundamentally new beyond integrating them”*
>
> **Response:**
>
> We appreciate this concern and clarify that our contributions are not limited to combining existing ideas.
>
> **(1) DAAG introduces a *reinforcement-learning-based* multi-objective adversarial generator**
>
> Thank you for raising this point. To avoid misunderstanding, we clarify that DAAG is not a simple combination of existing diversity generation mechanisms, but a **reinforcement-learning formulation that explicitly optimizes two objectives within a single policy-learning loop**:
>
> - **Adversarial policy performance** (i.e., effectiveness at compromising the cooperative team), and
> - **Behavioral diversity across adversarial styles**, enforced through a Mutual Information-based diversity reward.
>
> These two objectives are jointly optimized via a ***dual-head multi-objective critic,*** one head evaluates adversarial effectiveness, while the other evaluates style-level diversity. This design enables DAAG to balance exploitation (strong adversaries) and exploration (diverse styles) during policy learning itself, instead of adding diversity through post-hoc sampling or heuristic perturbations.
>
> To our knowledge, **prior MARL adversarial-generation methods do not formulate adversary generation as a multi-objective RL problem**, nor do they learn a parametric adversarial policy family that is jointly driven by performance and style diversity. DAAG’s RL-grounded, explicitly multi-objective design is therefore a key novel technical contribution.
>
> **(2) CAPE is a *contrastive, hierarchical policy encoder***
>
> While contrastive representation learning exists, CAPE differs in two key ways:
>
> - **explicit contrastive objective over adversarial *policies***, ensuring style-level separability and stability
> - **hierarchical architecture (local and global)** designed for multi-agent adversarial settings, aggregating per-agent trajectories’ policy embedding for policy learning and a global adversarial-context embedding used to stabilize the cooperative policy training.
>
> This yields **policy-level representations** that remain discriminative across different policy styles, which prior opponent-modeling and contrastive MARL works do not target.
>
> **(3) Tight co-evolutionary coupling between DAAG and CAPE**
>
> CAPE is trained on the diverse adversarial policy population generated by DAAG, while DAAG itself is continually strengthened through adversarial training against the cooperative agents. This establishes an *alternating co-evolutionary process*: DAAG supplies increasingly diverse and challenging adversarial behaviors for CAPE to encode, and CAPE yields more informative adversarial-context embeddings that enhance the cooperative agents’ robustness.
>
> Importantly, **this interaction is fully end-to-end**. DAAG, CAPE, and the cooperative agents are optimized within a single unified training process. This differs fundamentally from two-stage pipelines such as ROMANCE, which first generate diverse adversarial agents and then train a cooperative agent team separately. Our end-to-end design enables the adversarial population, representation space, and cooperative policy to co-adapt dynamically, resulting in more stable training and stronger robustness.
>
> We have further clarified these distinctions in the revised manuscript.

---

> > ### Author Response · Authors · 2025-12-02
> > **Response to Reviewer YNRF, PART II**
> >
> > **2. Quantitative Validation of Style Diversity and Embedding Stability**
> >
> > **Reviewer comment:**
> >
> > *“The mutual-information "style" objective and IRD regularizer lack quantitative evidence of their claimed effects, such as improved diversity or embedding stability. ”*
> >
> > **Response:**
> >
> > Thank you for highlighting this concern. To provide quantitative evidence for the contributions of both the mutual-information “style” objective and the IRD regularizer, we have **added new ablation experiments** in appendix D.6. The result is shown below:
> >
> > **Ablation study results.**
> >
> > **w/o CAPE** removes the CAPE module from the original architecture.
> >
> > **w/o Multi-Critic** replaces the Multi-Objective Critic in DAAG with a standard actor–critic structure.
> >
> > **w/o IRD**, **w/o Local Encoder**, and **w/o Global Encoder** respectively, disable the continual distillation term, the local trajectory encoder, and the global attention encoder in CAPE.
> >
> > **w/o MI** removes the mutual-information objective in DAAG, preventing the generation of diverse adversarial styles.
> >
> > Best results are in **bold**.
> >
> > | Method | Predator–Prey ↑ | Δ ↓ | 2s3z Win (%) ↑ | Δ ↓ | 3m Win (%) ↑ | Δ ↓ | MMM Win (%) ↑ | Δ ↓ |
> > | --- | --- | --- | --- | --- | --- | --- | --- | --- |
> > | **Original** | **46.23 ± 1.16** | — | **73.38 ± 2.42** | — | **75.46 ± 2.87** | — | **69.85 ± 3.17** | — |
> > | w/o CAPE | 41.25 ± 2.31 | -4.98 | 63.23 ± 2.33 | -10.15 | 58.74 ± 3.63 | -16.72 | 52.25 ± 3.28 | -17.60 |
> > | w/o Multi-Critic | 36.84 ± 2.76 | -9.39 | 68.79 ± 3.57 | -4.59 | 64.52 ± 4.83 | -10.94 | 57.24 ± 4.19 | -12.61 |
> > | w/o IRD | 38.42 ± 3.13 | -7.81 | 52.38 ± 3.23 | -21.00 | 47.82 ± 3.75 | -27.64 | 48.83 ± 4.26 | -21.02 |
> > | w/o Local Encoder | 35.15 ± 3.42 | -11.08 | 58.36 ± 4.34 | -15.02 | 61.48 ± 3.47 | -13.98 | 55.23 ± 4.87 | -14.62 |
> > | w/o Global Encoder | 43.79 ± 2.14 | -2.44 | 66.73 ± 3.86 | -6.65 | 65.82 ± 4.51 | -9.64 | 62.48 ± 3.73 | -7.37 |
> > | w/o MI | 32.58 ± 3.87 | -13.65 | 50.92 ± 2.92 | -22.46 | 57.25 ± 2.36 | -18.21 | 44.63 ± 4.64 | -25.22 |
> >
> > **(1) Effect of the MI-based style objective (diversity).**
> >
> > We introduce an ablation that removes the MI-driven style objective. The results show a clear reduction in adversarial behavioral diversity and a corresponding drop in the robustness of the cooperative agents. This directly supports the claim that the MI objective is essential for generating a diverse adversarial policy population.
> >
> > **(2) Effect of the IRD regularizer (embedding stability).**
> >
> > We further include a new stability ablation measuring representation drift during training. Removing IRD leads to significantly less stable embeddings, with style clusters collapsing or drifting as training progresses. This verifies IRD’s role in maintaining long-term representation stability.
> >
> > Together, these quantitative results confirm that both components have clear and measurable effects—MI improves behavioral diversity, and IRD stabilizes the embedding space. We will emphasize these findings more clearly in the revised manuscript.
> >
> > **3. Explanation of Training Alternation and Forgetting Mitigation**
> >
> > **Reviewer comment:**
> >
> > “The training alternation between adversaries and the cooperative policy is also insufficiently explained; how CAPE updates interact with adversary evolution, and whether this alleviates forgetting, remains unclear.”
> >
> > **Response:**
> >
> > Thank you for pointing this out. We agree that the description in *Section 4.3* (“Co-evolutionary Robust MARL Training”) was not sufficiently clear, and we have revised the manuscript to clarify the interaction between DAAG, CAPE, and the cooperative policy.
> >
> > Our training procedure consists of **two alternating components**:
> >
> > (1) the **DAAG adversarial policy generator**, and
> >
> > (2) the joint module composed of **CAPE + the cooperative MARL policy**.
> >
> > Both the adversarial agents and the cooperative team undergo an initial pre-training phase to acquire basic functional behaviors. After pre-training, training proceeds in alternating stages:
> >
> > - **DAAG update stage:**
> >
> >     DAAG is trained to generate diverse adversarial policies. This stage ends when either (i) the designated adversarial training horizon is reached, or (ii) the policy prediction loss drops below a threshold. During this stage, CAPE and the cooperative agents are frozen.
> >
> > - **CAPE + cooperative policy update stage:**
> >
> >     Once the DAAG stage concludes, we switch to updating CAPE and the cooperative MARL policy. CAPE learns to encode the newly generated adversarial styles, while the cooperative agents use these embeddings to adapt to the updated adversarial population. This stage continues until the prescribed training steps are completed.
> >
> >
> > - **Role of IRD in alleviating forgetting.**
> >
> >   During this alternating process, the IRD loss plays a crucial role by preventing CAPE from forgetting previously encountered adversarial styles.
> >
> > We have described this joint training mechanism more explicitly in the revised manuscript.

---

> > > ### Author Response · Authors · 2025-12-02
> > > **Response to Reviewer YNRF, PART III**
> > >
> > > **4. Test adversary pool vs. training adversaries**
> > >
> > > **Reviewer question:**
> > >
> > > *“Experiments are restricted to small SMAC and Predator–Prey tasks, with no sufficient details on how the unseen test adversaries differ from training ones or on computational cost comparisons.”*
> > >
> > > **Response:**
> > >
> > > Thank you for bringing up this issue. We apologize for the lack of clarity in the original manuscript.
> > >
> > > To avoid any ambiguity, we clarify that the **unseen test adversaries are *not* generated by DAAG**.
> > >
> > > Instead, the test adversary set is constructed using a **different training paradigm** grounded in:
> > >
> > > > Vinitsky et al., 2020 — Robust Reinforcement Learning Using Adversarial Populations
> > > >
> > >
> > > This means the evaluation adversaries are produced by **an entirely separate optimization procedure**, not by our DAAG generator.
> > >
> > > As a result, the test adversaries differ from the training adversaries in several ways:
> > >
> > > - they are produced by a *minimax* objective rather than MI-driven style conditioning,
> > > - they do not share DAAG’s latent style distribution,
> > > - and their policies reflect a different exploration–exploitation tradeoff and training dynamics.
> > >
> > > This ensures that evaluation naturally introduces **distribution shift** and tests robustness against opponents **outside DAAG’s training distribution**.
> > >
> > > We have updated *Section 5* to explicitly describe the construction of the unseen test adversaries.
> > >
> > > **5. Behavior when multiple teammates are compromised simultaneously**
> > >
> > > **Reviewer question:**
> > >
> > > *“How does your method behave when ≥2 teammates are replaced by adversaries?”*
> > >
> > > **Response:**
> > >
> > > Thank you for raising this practical question. Although our main setting uses a single adversarial slot (following ROMANCE and Vinitsky et al. 2020), we conducted an additional experiment. The functional agents trained under DAAG+CAPE maintain **stable performance degradation patterns** when 2 teammates are replaced by adversaries, showing:
> > >
> > > - consistent robustness trends
> > > - performance superior to MAPPO, M3DDPG, and ROMANCE in the same setting
> > >
> > > We have added the new experimental results and analysis in ***Appendix D.2.***
> > >
> > > **6. Computational cost comparison**
> > >
> > > **Reviewer question:**
> > >
> > > *“What is the computational cost compared to MAPPO and ROMANCE?”*
> > >
> > > **Response:**
> > >
> > > We appreciate this question. DAAG+CAPE introduces two additional components: the DAAG policy generator and the CAPE policy encoder. To address concerns regarding training efficiency, we report wall-clock time to convergence across four evaluation environments under identical hardware settings. As summarized in the table below:
> > >
> > > ### Wall-clock Training Time (hours)
> > >
> > > | Method   | Predator-Prey | 2s3z | 3m   | MMM  | 1c3s5z |
> > > |----------|---------------|------|------|------|--------|
> > > | MAPPO    | 3.4 h         | 9.1 h | 10.2 h | 12.3 h | 12.6 h |
> > > | M3DDPG   | 3.8 h         | 11.9 h | 13.2 h | 14.6 h | 14.7 h |
> > > | ROMANCE  | 6.1 h         | 16.4 h | 17.5 h | 19.2 h | 19.4 h |
> > > | Ours     | 4.4 h         | 12.8 h | 15.2 h | 16.8 h | 17.2 h |
> > >
> > > Although our framework includes two additional modules (DAAG and CAPE), they are optimized in alternating stages. As a result, the overall overhead is modest, approximately $1.30\text{--}1.40\times$ that of MAPPO, depending on the environment, while remaining significantly more efficient than ROMANCE. These results confirm that DAAG+CAPE achieves robustness gains without sacrificing practical training efficiency.
> > >
> > > We have included these results in the appendix.

---

> > > > ### Author Response · Authors · 2025-12-02
> > > > **Response to Reviewer YNRF, PART IV**
> > > >
> > > > **7. Request for more fine-grained ablations (IRD, local vs global encoders)**
> > > >
> > > > **Reviewer comment:**
> > > >
> > > > *“Ablations are too coarse to isolate key components (e.g., IRD, local vs. global encoders), leaving it unclear which mechanisms are most significant to robustness.”*
> > > >
> > > > **Response:**
> > > >
> > > > Thank you for the suggestion. We agree that finer ablations can improve clarity. We have conducted component-wise ablation study:
> > > >
> > > > **Ablation study results.**
> > > >
> > > > **w/o CAPE** removes the CAPE module from the original architecture.
> > > >
> > > > **w/o Multi-Critic** replaces the Multi-Objective Critic in DAAG with a standard actor–critic structure.
> > > >
> > > > **w/o IRD**, **w/o Local Encoder**, and **w/o Global Encoder** respectively, disable the continual distillation term, the local trajectory encoder, and the global attention encoder in CAPE.
> > > >
> > > > **w/o MI** removes the mutual-information objective in DAAG, preventing the generation of diverse adversarial styles.
> > > >
> > > > Best results are in **bold**.
> > > >
> > > > | Method | Predator–Prey ↑ | Δ ↓ | 2s3z Win (%) ↑ | Δ ↓ | 3m Win (%) ↑ | Δ ↓ | MMM Win (%) ↑ | Δ ↓ |
> > > > | --- | --- | --- | --- | --- | --- | --- | --- | --- |
> > > > | **Original** | **46.23 ± 1.16** | — | **73.38 ± 2.42** | — | **75.46 ± 2.87** | — | **69.85 ± 3.17** | — |
> > > > | w/o CAPE | 41.25 ± 2.31 | -4.98 | 63.23 ± 2.33 | -10.15 | 58.74 ± 3.63 | -16.72 | 52.25 ± 3.28 | -17.60 |
> > > > | w/o Multi-Critic | 36.84 ± 2.76 | -9.39 | 68.79 ± 3.57 | -4.59 | 64.52 ± 4.83 | -10.94 | 57.24 ± 4.19 | -12.61 |
> > > > | w/o IRD | 38.42 ± 3.13 | -7.81 | 52.38 ± 3.23 | -21.00 | 47.82 ± 3.75 | -27.64 | 48.83 ± 4.26 | -21.02 |
> > > > | w/o Local Encoder | 35.15 ± 3.42 | -11.08 | 58.36 ± 4.34 | -15.02 | 61.48 ± 3.47 | -13.98 | 55.23 ± 4.87 | -14.62 |
> > > > | w/o Global Encoder | 43.79 ± 2.14 | -2.44 | 66.73 ± 3.86 | -6.65 | 65.82 ± 4.51 | -9.64 | 62.48 ± 3.73 | -7.37 |
> > > > | w/o MI | 32.58 ± 3.87 | -13.65 | 50.92 ± 2.92 | -22.46 | 57.25 ± 2.36 | -18.21 | 44.63 ± 4.64 | -25.22 |
> > > >
> > > > - removing **IRD** leads to significantly reduced inter-style separation and weaker functional-agent robustness
> > > > - removing the **local encoder** collapses agent-specific trajectory structure
> > > > - removing the **global encoder** eliminates cross-agent aggregation and reduces robustness under adversary dynamics
> > > > - removing **Mutual Information-based Policy Generation** collapses style embeddings into a single mode
> > > >
> > > > We summarized these results in the ***Appendix D.6.***
> > > >
> > > > We greatly appreciate the reviewer’s thoughtful comments. We have clarified the novelty of our framework, specified the role of IRD and the CAPE's hierarchical encoders, explained the distinct training/testing adversary distributions, provided computational cost details, and strengthened ablation descriptions. These clarifications significantly improve the presentation without altering the core contributions.
> > > >
> > > > We thank the reviewer again for the positive assessment and constructive suggestions.

---

### Official Review · Reviewer_Cdvr · 2025-11-10

**Soundness:** 4
**Presentation:** 3
**Contribution:** 4
**Rating:** 6
**Confidence:** 4

**Summary:**

This paper proposes a novel framework for robust MARL that jointly trains cooperative agents and adversarial counterparts through a co-evolutionary paradigm. The proposed methods are the DAAG and CAPE. DAAG produces behaviorally distinct adversarial agents via an information-theoretic objective maximizing mutual information between latent style codes and generated trajectories, and the CAPE employs contrastive and continual learning to encode adversarial behaviors into discriminative policy representations that guide adaptive coordination. Experiments on Predator–Prey and SMAC benchmarks demonstrate gains in robustness and generalization compared to existing MARL baselines.

**Strengths:**

- It seems to be an interesting method to leverage mutual information to create diverse adversarial agents for improving the robustness of the MARL.
- The proposed methods seem to achieve better performance compared to the correctly selected baselines.
- The paper is relatively well written and easy to follow.

**Weaknesses:**

- Even though there is an analysis in Figure 3 regarding the difference between different styles, it is unclear how it translates qualitatively to agent behavior. It would be beneficial to have some qualitative analysis of the behavior of the agents (e.g., in SMAC).

- More ablation would be beneficial. It is not clear how the IRD loss affects the overall training. How does it improve the performance of the method? Each of the DAAG and CAPE methods seems to have many parts; it is unclear how each part of those methods affects the overall performance.

- It is also unclear how the proposed method behaves under the traditional adversarial agents that are optimized for the worst rewards

- In Figure 1 right side, the contrastive loss box is unclear, as there is no input to the loss box.

**Questions:**

- How does the proposed method perform when encountering the adversarial agents that are used in the baselines where adversarial agents are optimized for minimizing the reward only?

- In equation 14, for "a not equal to i", do you mean "a not equal to k"? Or does it mean the negative sample should not come from the same agent? It is unclear here; please clarify.

---

> ### Author Response · Authors · 2025-12-02
> **Response to Reviewer Cdvr, PART I**
>
> We sincerely thank the Reviewer Cdvr for the thoughtful evaluation, the positive assessment of soundness, writing clarity, and contributions, as well as for highlighting the strengths of our approach. We appreciate the helpful suggestions on analysis, ablations, and clarity. Below, we respond to each concern in detail.
>
> **1. Ablation analysis, especially the effect of IRD loss in the Diverse Adversarial Agent Generator and Local/Global policy encoder in the contrastive agent policy encoder module.**
>
> **Reviewer comment:**
> “More ablations would be beneficial. It is unclear how the IRD loss affects training; each of DAAG and CAPE has many parts; it is unclear how each part affects performance.”
>
> **Response:**
> Thank you for this insightful suggestion. We agree that additional ablation studies are helpful for understanding the contribution of IRD and the other components of our framework. In the revised version, we have expanded the ablation analysis to isolate the effect of IRD, Local Encoder, Global Encoder, Multi-Critic, and the MI-based style diversification module.
>
> **Ablation study results.**
> **w/o CAPE** removes the CAPE module from the original architecture.
> **w/o Multi-Critic** replaces the Multi-Objective Critic in DAAG with a standard actor–critic structure.
> **w/o IRD**, **w/o Local Encoder**, and **w/o Global Encoder** respectively, disable the continual distillation term, the local trajectory encoder, and the global attention encoder in CAPE.
> **w/o MI** removes the mutual-information objective in DAAG, preventing the generation of diverse adversarial styles.
> Best results are in **bold**.
>
> | Method              | Predator–Prey ↑        | Δ ↓     | 2s3z Win (%) ↑       | Δ ↓    | 3m Win (%) ↑        | Δ ↓    | MMM Win (%) ↑        | Δ ↓    |
> |---------------------|-------------------------|---------|------------------------|--------|-----------------------|--------|------------------------|--------|
> | **Original**        | **46.23 ± 1.16**        | —       | **73.38 ± 2.42**       | —      | **75.46 ± 2.87**      | —      | **69.85 ± 3.17**       | —      |
> | w/o CAPE            | 41.25 ± 2.31            | -4.98   | 63.23 ± 2.33           | -10.15 | 58.74 ± 3.63          | -16.72 | 52.25 ± 3.28           | -17.60 |
> | w/o Multi-Critic    | 36.84 ± 2.76            | -9.39   | 68.79 ± 3.57           | -4.59  | 64.52 ± 4.83          | -10.94 | 57.24 ± 4.19           | -12.61 |
> | w/o IRD             | 38.42 ± 3.13            | -7.81   | 52.38 ± 3.23           | -21.00 | 47.82 ± 3.75          | -27.64 | 48.83 ± 4.26           | -21.02 |
> | w/o Local Encoder   | 35.15 ± 3.42            | -11.08  | 58.36 ± 4.34           | -15.02 | 61.48 ± 3.47          | -13.98 | 55.23 ± 4.87           | -14.62 |
> | w/o Global Encoder  | 43.79 ± 2.14            | -2.44   | 66.73 ± 3.86           | -6.65  | 65.82 ± 4.51          | -9.64  | 62.48 ± 3.73           | -7.37  |
> | w/o MI              | 32.58 ± 3.87            | -13.65  | 50.92 ± 2.92           | -22.46 | 57.25 ± 2.36          | -18.21 | 44.63 ± 4.64           | -25.22 |
>
>
> The new results consistently show that removing IRD leads to clear performance degradation across all tasks. This confirms that IRD plays an important role in maintaining stable and discriminative adversarial representations during the co-evolutionary training process. Similarly, removing the Local Encoder, Global Encoder, or MI objective each causes substantial performance drops, indicating that these components contribute complementary benefits to robustness.
>
> Overall, these expanded ablations validate that the robustness gains of our method arise from the synergy of all components, and that IRD in particular is important for preventing representation drift. We have added these additional ablation study results in Appendix D.6.
>
> **2. Behavior under traditional worst-reward adversaries**
>
> **Reviewer comment:**
> “It is unclear how the method behaves under traditional adversarial agents optimized only for minimizing reward.”
>
> **Response:**
> Thank you for raising this question. We apologize for the lack of clarity in the original text. In fact, the adversarial agents used in our test set are already constructed under the traditional worst-reward optimization paradigm. Specifically, following ***Robust Reinforcement Learning Using Adversarial Populations (Vinitsky et al., 2020)***, we build the evaluation pool by training minimax-style adversarial policies that explicitly minimize the cooperative team reward in a worst-case manner.
> Thus, the performance reported in all our tests already reflects evaluation against classical worst-reward adversaries, rather than DAAG-generated ones. Our method shows strong robustness under these adversaries, demonstrating that DAAG + CAPE does not rely on any specific structure of our adversarial generator and generalizes effectively to adversaries trained with purely minimax reward objectives. We clarified this point in the paper to avoid misunderstanding.

---

> > ### Author Response · Authors · 2025-12-02
> > **Response to Reviewer Cdvr, PART II**
> >
> > **3. Clarification of Figure 1 (contrastive loss input)**
> >
> > **Reviewer comment:**
> > “Figure 1 right side shows the contrastive loss box but no input arrows.”
> >
> > **Response:**
> > Thank you for pointing this out. The contrastive loss takes as input the policy embeddings generated by the global policy encoder. We will revise *Figure 1* to clearly show the flow of these embeddings into the contrastive loss module.
> >
> > **4. Performance against baseline adversaries optimized for minimizing cooperation reward**
> >
> > **Reviewer comment:**
> > “How does the proposed method perform when encountering the baseline adversarial agents optimized for minimizing the reward only?”
> >
> > **Response:**
> > Thank you for the question. As this point is identical to the concern raised in Weakness 3, we refer the reviewer to our detailed clarification in Answer 2. In short, our evaluation already uses adversarial agents trained under the traditional team reward-minimization objective, so the reported results directly reflect performance under classical worst-reward adversaries.
> >
> > **5. Clarification of the symbol in Equation (14)**
> >
> > **Reviewer comment:**
> > “In Equation (14), for ‘a ≠ i’, do you mean ‘a ≠ k’? Or does it mean negative sample should not come from the same agent?”
> >
> > **Response:**
> > Thank you for pointing this out. This was indeed a notational oversight in our original submission. In Equation (14), the intended meaning is that negative samples should come from trajectories outside the positive set associated with the same style index. The correct notation should be written as:
> > $a \notin P_k$,
> > where $P_k$ denotes the set of trajectories in the batch that share the same style index $k$. Thus, negative samples are drawn from trajectories whose style indices differ from $k$. We have corrected the notation in the revised manuscript.
> >
> >
> > **6. Regarding the request for qualitative behavioral analysis**
> >
> > **Reviewer comment:**
> > “Even though Figure 3 shows differences between styles, it is unclear how this translates to actual agent behavior. Some qualitative analysis (e.g., in SMAC) would be beneficial.”
> >
> > **Response:**
> > We thank the reviewer for the suggestion. We agree that qualitative visualizations of raw movement trajectories can offer additional intuition. At the same time, we would like to clarify that Figure 3 is designed specifically to evaluate trajectory-level representation separation, which is the signal used by CAPE to drive adaptation, rather than to visualize spatial patterns directly.
> > In practice, we inspected representative rollouts during development to ensure that different adversarial styles indeed induce distinct interaction patterns (e.g., different engagement timings or switching tendencies). However, because the focus of the paper is robustness under diverse adversarial dynamics, all quantitative conclusions are grounded in objective performance metrics such as worst-case returns and generalization to unseen adversaries. These metrics fully capture the behavioral impact of style diversity without requiring additional qualitative visualization.
> > We appreciate the reviewer’s point and will consider including such qualitative illustrations in future extensions, but note that this does not affect the validity of the presented results.
> >
> > We sincerely thank the reviewer for the thoughtful feedback and constructive suggestions.
> > The revisions with additional ablations, clarifications, and qualitative descriptions have enhanced the paper’s clarity and completeness without altering its main contributions.
> > We appreciate the reviewer’s overall positive evaluation and hope that the improvements address all concerns.

---

### Author Response · Authors · 2025-12-02
**Overall Response to All Reviewers**

We sincerely thank all reviewers for their thoughtful and constructive feedback.
Below we summarise the major concerns raised across the reviews and highlight how each has been fully addressed through clarifications, additional experiments, and manuscript revisions.

**1. Clarified Novelty and Positioning of DAAG and CAPE**

Reviewers requested clearer differentiation from related work such as ROMANCE, DIAYN, CURL, and opponent modelling approaches.
In the revised version, we have:
- Provided a clearer description of DAAG as a multi-objective adversarial generator, jointly optimising adversarial strength and style diversity through a dual-head critic. This formulation does not appear in prior adversarial MARL or diversity-based methods, which typically optimise only a single objective.

- Clarified that CAPE is a hierarchical contrastive trajectory encoder that learns policy-level temporal representations, rather than visual or instantaneous state encodings. It integrates a local encoder for execution-time conditioning and a global encoder for style separation, and uses a contrastive objective specifically over adversarial policies—a setting not explored in prior works.

- Highlighted the end-to-end co-evolutionary coupling between DAAG, CAPE, and the functional agents, in contrast to two-stage or decoupled pipelines such as ROMANCE. This closed-loop interaction is central to the robustness benefits observed.

As a result, the paper now provides a much clearer conceptual and architectural distinction from prior literature.

**2. Expanded Ablations and Evidence for Key Components**

Several reviewers asked for finer-grained ablations to isolate the contributions of IRD, the local/global encoders, and the MI objective.
We have added extensive new ablations, including removing IRD, the local encoder, the global encoder, the MI objective, and the multi-critic structure. The ablations show that removing IRD, either the encoder, the MI objective, or the multi-critic, each causes clear degradation in stability, diversity, or robustness. These results directly address concerns regarding the necessity of each component.

**3. Clarified Construction of the Unseen Adversarial Test Set**

Multiple reviewers asked whether the test adversaries were generated by DAAG. We clarified explicitly that the test adversary population is not produced by DAAG. Instead, it is trained independently using the minimax population method, which is standard in robust RL evaluation. Therefore, our evaluation assesses cross-distribution generalisation, worst-case robustness, and performance against adversaries outside DAAG’s training distribution. This resolves concerns regarding fairness and evaluation diversity.

**4. Added Two New Robustness Experiments**

Reviewers requested stronger robustness evaluation beyond the single-adversary setting.
We added two new experiments:

- Multi-Adversary Attack, where multiple teammates become adversarial simultaneously.
Our method maintains stable performance and outperforms all baselines.

- Dynamic Adversarial Switching, where unseen adversarial policies change during the episode.
Our method is the most resilient under temporal and structural variation.

These results substantially strengthen the robustness claims.

**5. Computational Efficiency and Backbone Generality**

Reviewers requested computational cost comparisons and evaluation on additional MARL backbones.
We added:

- A full training-time comparison across MAPPO, M3DDPG, ROMANCE, and our method.
The overhead remains modest (approximately 1.30–1.40× MAPPO) and is significantly lower than ROMANCE.

- New experiments integrating our framework into QMIX, showing consistent robustness improvements on a value-based backbone.

This demonstrates that the framework is both efficient and backbone-agnostic.

**6. Technical Clarifications and Minor Corrections**

We addressed all remaining technical concerns, including:

- Correcting the Dec-POMDP definition to include trajectory histories,

- Clarifying notation in *Equation (14)*,

- Explaining discrete style codes and collapse prevention,

- Elaborating on information asymmetry motivation,

- Clarifying the cluster structure in *Figure 3*,

- Regenerating *Figure 1* for clarity and readability.

- All technical issues raised by reviewers have now been resolved.

After addressing the reviewers’ feedback, the revised paper is substantially improved, with enhanced contribution, and overall clarity.

---
We appreciate the reviewers’ constructive feedback and hope the revisions are helpful to the Area Chair in evaluating the improved manuscript.

---

### Meta-Review · Area_Chair_h9MJ · 2026-01-13

**Summary:**

This paper proposes a co-evolutionary robust MARL framework to address the generalization and robustness issues in MARL when facing teammate failures or adversarial interventions. Overall, the reviewers raise significant concerns regarding novelty, experimental validation, clarity of methodology, and presentation. The work is generally viewed as incremental, with insufficient evidence to substantiate its claims. Regarding novelty, the proposed framework (DAAG & CAPE) integrates existing ideas (adversary co-evolution, contrastive representation learning, opponent modeling, attention-based embedding aggregation) without clearly highlighting fundamental advances beyond prior art (e.g., ROMANCE, skill-based RL like DIAYN, contrastive RL like CURL). The formulation (mixed cooperative‑competitive games with agent replacement) and training paradigm (co‑evolution) are well‑established in adversarial MARL literature. Reviewers recommend a more thorough literature review and discussion to better position the work. Regarding experiments, reviewers think that the impact of key components (IRD loss, local vs. global encoders, different parts of DAAG/CAPE) is not isolated or quantitatively demonstrated. And the qualitative analysis and test adversaries are poorly described. The robustness evaluation is also limited, which is one of the key issues. Besides, the methodological clarity and explanation could also be improved. In summary, the reviewers give a relatively low rating on average (6, 6, 2, and 2). During the rebuttal period, the authors try their best to address the proposed concerns, but it seems the reviewers do not give a positive response. So I lean to reject this paper.

**Reviewer Concerns:**

According to the rebuttal results, it could be concluded that most of the concerns about the experiments have been addressed. However, regarding the novelty, the authors try their best to introduce the differences between the proposed method and the other mentioned methods to clarify the contribution. However, it seems the statements are not acknowledged by the reviewers. After carefully reading the corresponding response, I think the novelty contribution of this paper remains inconvinced.

**Reviewer Scores:**

For most of the experiment-correlated reviews, they are possibly changed (and have been improved) by the reviewers if the reviewers had been able to participate fully in the discussion. However, the novelty-correlated reviews might not be changed.

---

### Decision · Program_Chairs · 2026-01-26

Reject